# A voxel-based approach for simulating microbial decomposition in soil: Comparison with LBM and improvement of morphological models

Mouad Klai[1,2], Olivier Monga[1,2], Mohamed Soufiane Jouini[3]*, Valérie Pot[4]

**1** Laboratory of Mathematics and Population Dynamics (LMDP), Cadi Ayyad University, Marrakech, Morocco, **2** Unit for Mathematical and Computer Modeling of Complex Systems (UMMISCO), IRD, Sorbonne University, Paris, France, **3** Mathematics Department, Khalifa University of Science and Technology, Abu Dhabi, United Arab Emirates, **4** Université Paris-Saclay, INRAE, AgroParisTech, UMR ECOSYS, Palaiseau, France

* mohamed.jouini@ku.ac.ae

## Abstract

This paper deals with the computational modeling of biological dynamics in soil using an exact micro-scale pore space description from 3D Computed Tomography (CT) images. Within this context, computational costs and storage requirements constitute critical factors for running simulations on large datasets over extended periods. In this research, we represent the pore space by a graph of voxels (Voxel Graph-Based Approach, VGA) and model transport in fully saturated conditions (two-phase system) using Fick's law and coupled diffusion with biodegradation processes to simulate microbial decomposition in soil. To significantly decrease the computational time of our approach, the diffusion model is solved by means of Euler discretization schemes, along with parallelization strategies. We also tested several numerical strategies, including implicit, explicit, synchronous, and asynchronous schemes. To validate our VGA, we compare it with LBioS, a 3D model that integrates diffusion (via the Lattice Boltzmann method) with biodegradation, and Mosaic, a Pore Network Geometrical Modelling (PNGM) which represents the pore space using geometrical primitives. Our method yields result similar to those of LBioS in a quarter of the computing time. While slower than Mosaic, it is more accurate and requires no calibration. Additionally, we show that our approach can improve PNGM-based simulations by using a machine-learning approach to approximate diffusional conductance coefficients.

## 1. Introduction

Microbial activity in soil is essential for maintaining soil health and supporting ecosystem functioning. Microorganisms contribute to nutrient cycling, organic matter decomposition, and plant growth promotion [1–5]. However, soil is a highly diverse and dynamic environment, making it challenging to capture its full complexity. Additionally, as a heterogeneous and intricate medium, it poses difficulties for directly observing and measuring microbial activity [3]. Traditional approaches, such as culturing and microscopy, are time-consuming

**Data availability statement:** All relevant data are within the manuscript and its Supporting Information files we have included a link (https://github.com/mouadklai/VGA_microbial_decomposition) to the C implementation for further reference.

**Funding:** The author(s) received no specific funding for this work.

**Competing interests:** The authors have declared that no competing interests exist.

and provide limited insights. Therefore, innovative and advanced numerical simulation techniques are needed to study soil microbial activity [4–9].

Computed Tomography (CT) imagery offers a non-invasive method for visualizing porous media structures at the microscale of the soil microbial habitats. Standard 3D CT imaging generates large image data volumes, often around one billion voxels [10–12]. Over the past two decades, significant research has focused on improving computational models for simulating dynamics from 3D CT images of soil. These works aimed to better understand the movement and interaction of soil components in porous and fractured media. These studies primarily integrate transport mechanisms with advanced reaction (transformation) processes [13–16].

Several methodologies have been proposed to simulate chemical transport phenomena in soil, including the Lattice-Boltzmann Method (LBM) [8], smooth particle hydrodynamics [14], hybrid Lattice Boltzmann-direct numerical simulation (DNS) [15], and pore network geometrical models (PNGM) [16]. Among these, LBM-based approaches have gained popularity for simulating diffusion in porous media using micro 3D CT soil images [17–19]. However, LBM-based models are computationally expensive due to their particle-based process of collision and propagation, leading to high memory requirements and huge computing times. For example, on a regular laptop, a five-day real-time simulation of microbial decomposition of organic matter with an image size of $512 \times 512 \times 512$ voxels and a porosity of 17% takes approximately three weeks [20].

In contrast, PNGM-based simulations offer a cost-effective alternative with reduced computing time and moderate memory requirements. These models represent pore space as a graph of connected primitives, such as spheres, ellipsoids, or cylinders [20–23]. However, PNGM simulations require calibration, particularly in optimizing transport (diffusion) by adjusting diffusional conductance coefficients between connected primitives [24–26]. Despite their computational efficiency, PNGM models are limited by their dependence on such calibrations and may lack precision due to the difference in size distribution of the geometrical primitives used.

Zech et al. (2022) simulated microbial decomposition in porous media by coupling diffusion with a microbial decomposition model within a partial differential equation framework, solved using the Galerkin method. Similar to our approach, they numerically solved a PDE-based reaction-diffusion system within pore spaces derived from CT imaging [27]. However, this approach is limited to 2D due to the high computational cost of extending it to 3D. This highlights the ongoing need for scalable 3D solutions that combine the accuracy of voxel-based models with efficient computational strategies. This study addresses key limitations in existing methods by introducing a new computational approach for simulating microbial activity in soil using 3D CT images of pore space. Unlike traditional graph-based techniques modeling pore space with geometrical primitives, our method directly represents the pore space as a graph of connected voxels and models transport phenomena using Fick's law of diffusion [28]. The resulting improved balance obtained between accuracy and complexity makes it possible to generate data for learning diffusional conductance coefficients in pore network geometrical models.

In the Materials and Methods section, we describe the coupling of diffusion processes with a validated model for microbial activity, based on the SOMKO [30] and MIOR [31] models, where microbes undergo survival, maintenance, and growth. We detail the used Euler discretization schemes (Sections 2.1-2.4). We then present the calculation of diffusional conductance coefficients in pore network geometrical models (Section 2.5). In Section 2.6 we describe the data set used to compare our new VGA to LBioS and PNGM. We present and discuss the results in Section 3.

## 2. Materials and methods

### 2.1 Simulation principles

In this study, we only considered the condition that soil is fully saturated, i.e., soil pore space is filled with water.

We model the dynamics of microbial decomposition using five key compounds, as illustrated in Fig 1:

- Microbial biomass (MB): Represents the mass of microorganisms in the sample.

- Microbial respiration ($CO_2$): Indicates the produced carbon dioxide through microbial decomposition, indicative of microbial growth.

- Fresh organic matter (FOM): Derived from recently added or deposited plant and easily decomposable

- Soil organic matter (SOM): Consists of various organic compounds in different stages of decomposition, originating from biomass turnover and less accessible to decomposition.

- Dissolved organic matter (DOM): Refers to organic compounds dissolved in soil water, originating from the hydrolysis of FOM and SOM and biomass recycling, available for microbial uptake or transport within the pore space.

Note again that for environments where saturation is not complete, the same simulation framework can be used after a method of drainage [17,18] which refers to the method used to redistribute liquid and air within a porous medium, typically by gradually removing liquid from the system. This process mimics the movement of liquid due to capillary forces and gravity, leading to the formation of air-filled pores. In simulations, this is often achieved by progressively reducing the liquid content and adjusting the pore space accordingly. This process allows the model to represent partially saturated conditions before transforming the system into a two-phase configuration by treating the air phase as a non-interacting solid phase [17,18].

Let $\left(I(i,j,k)\right)_{i,j,k}$ be a 3D binary image where the voxels forming pore space (void voxels) are tagged by 0 and the voxels attached to solid matter (solid voxels) are tagged by 1. Let $V = \left\{(i,j,k): I(i,j,k)=0\right\} = \left\{v_1,\dots,v_n\right\}$ be the set of pore space voxels and $N = \left\{1,\dots,n\right\}$ the index set of $V$, where $n = card(V)$ represents the number of voxels of the pore space.

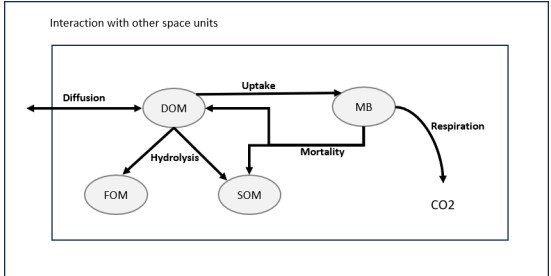

**Fig 1. The compounds and processes involved within the microbial decomposition of organic matter in soil: DOM comes from the decomposition of SOM (slow decomposition) and FOM (fast decomposition).** The microorganisms grow by assimilating DOM, breathe by producing CO2 and when they die, they are recycled into DOM and SOM.

We construct an adjacency valuated graph $G(V,E)$ where $V$ is the set of nodes, and $E = \{(i,j) \in N^2 | v_i \cap v_j \neq \varnothing\}$ is the set of edges. In this work, we use 6-connecvity between voxels (Fig 2).

Let $t$ be the variable representing time. To each voxel $v_i$ of the pore space, we attach: the mass of MB ( $x_i^1(t)$ ), the mass of DOM ( $x_i^2(t)$ ), the mass of SOM $\left(x\|i^3(t)\right)$, the mass of FOM ( $x_i^4(t)$ ), the mass of $CO_2$ $\left(x\|i^5(t)\right)$.

To each node $v_i$ let us denote by $X_i(t) = \left(x_i^1(t), x_i^2(t), x_i^3(t), x_i^4(t), x_i^5(t)\right)$ the masses of the compounds contained within voxel $v_i$ at time t.

We simulate microbial activity by updating the constructed graph according to transformation and diffusion processes described in Fig 3. The principle is to break down the complex processes, that can be mathematically modelled, into simpler steps.

Let us assume that we get a representation of the pore space and its compounds at time $t$ in the form of $G_t\left(V, E, \left(X_i(t)\right)_{i \in N}\right)$.

To get $G_{t+\delta t}\left(V, E, \left(X_i(t+\delta t)\right)_{i \in N}\right)$ at time $t + \delta t$, after microbial activity processes we apply a transformation model that encodes the biological conversion of different compounds within each voxel. Next, we employ a discretized model to mimic the diffusion of compounds between the connected voxels. The modeling approach is based on dynamics splitting approach, where the diffusion and reaction processes are handled separately.

The spatial discretization is fixed according to the computed tomography image resolution. The same framework can be used to model diffusion-transformation processes at any

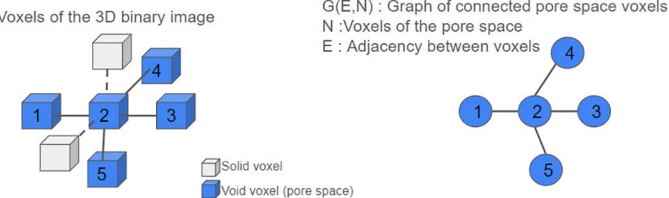

**Fig 2. Voxel representation of a 3D binary image (left) and its corresponding graph of connected pore space voxels (right).** The blue cubes represent void voxels (pore space), while the white cubes represent solid voxels. The graph shows the adjacency of the void voxels (pore space), where node 2 is connected to nodes 1, 3, 4, and 5, indicating geometrical proximity in the 3D space.

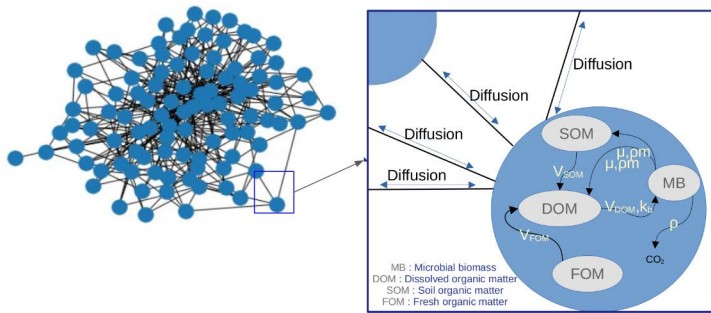

**Fig 3. Graph representation of the microbial activity model.** Each node represents a spatial unit, with links indicating geometrical adjacency between nodes. Microbial processes are split into diffusion and transformation, where transformation occurs within nodes and diffusion is modeled as mass exchange between connected nodes according to Fick's law.

resolution, as long as the parameters are converted based on the voxel size, which depends on the resolution of the 3D image.

## 2.2 Transport modelling and graph diffusion equation

**2.2.1 Fick's law of diffusion.** Fick's law of diffusion describes the process of molecular or particle movement through a medium, such as gases, liquids, or solids. It is expressed as follows [27]:

$$J = -D\frac{dC}{dx} \tag{1}$$

Where $J$ represents the diffusion flux (amount of substance per unit area per unit time), $D$ is the of the substance and $dc/dx$ the concentration gradient along the diffusion direction.

**2.2.2 Diffusion modelling: graph diffusion equation.** We consider the diffusion of dissolved organic matter (DOM). Let $m_i(t) = x_i^2(t)$ represent the mass of DOM at time $t$ in the voxel $p_i$. The diffusion process in our framework is modeled using a graph-based approach where voxels represent graph nodes, and edges connect neighboring voxels. The concentration at a voxel corresponds to its mass since the voxel volume is normalized.

The mass flow between time $t$ and time $t + \delta t$ from the voxel at position $p_i$ to its neighboring voxel at position $p_i$ is given by the first Fick's law as follows:

$$J_{i,j} = -D_{DOM}.\big(m_i(t) - m_j(t)\big).\delta t \tag{2}$$

where $D_{DOM}$ is the DOM diffusion coefficient in water.

Thus, the DOM mass at voxel i at time $t + \delta t$ due to mass exchange with neighbouring voxels is expressed as follows:

$$m_i(t + \delta t) - m_i(t) = \sum_{j:(i,j)\in E} -D_{DOM}\delta m_{i,j}(t)\delta t \tag{3}$$

Then the governing equation is:

$$\frac{dm_i(t)}{dt} = \sum_{j:(i,j)\in E} -D_{DOM}\big(m_i(t) - m_j(t)\big) \tag{4}$$

Gathering all the masses of all the voxels of the pore space in one vector $M(t) = \big(m_1(t)...m_n(t)\big)$ we get

$$\frac{dM(t)}{dt} = -D_{DOM}.\Delta M(t) \tag{5}$$

Where $\Delta = \big(\delta_{i,j}\big)_{1\leq i,j\leq n}$ is the Laplacian matrix of the graph $G(E,N)$ defined by:

$$\left\{ \begin{array}{ll} \delta_{i,j} = deg(i) & if i = j \\ \delta_{i,j} = -1 & if i \neq j \wedge (i,j) \in E \\ \delta_{i,j} = 0 & otherwise \end{array} \right.$$

where $deg(i) = \sum_{j:(i,j)\in E} 1$ is the number of adjacent nodes to the node i.

Equation (5) is called Graph Diffusion Equation (GDE) of $G(N,E)$.

Numerical schemes

For temporal discretization, we employed two different schemes:

- Explicit scheme (forward Euler)

$$m_i^{(k+1)} = m_i^{(k)} - \sum_{j:(i,j) \in E} D_{DOM} \delta t \left( m_i^{(k)} - m_j^{(k)} \right) \tag{6}$$

- Implicit scheme (backward Euler).

$$m_i^{(k)} = m_i^{(k+1)} + \sum_{j:(i,j) \in E} D_{DOM} \delta t \left( m_i^{(k+1)} - m_j^{(k+1)} \right) \tag{7}$$

The explicit scheme (Eq. 6) offers simplicity but may suffer from stability issues, while the implicit scheme (Eq. 7), solved using the conjugate gradient method, provides robustness at the cost of computational complexity. The detailed formulations of the numerical schemes are provided in the supplementary material S1 File. where $k$ is the iteration number (i.e., the discrete-time index), and $\delta t$ is the time step.

## 2.3 Modelling microbial decomposition of organic matter processes in the graph of connected voxels

Microbial activity is governed not only by the diffusion of various compounds within the soil's pores but also by transformation processes. Dissolved Organic Matter (DOM) arises from the decomposition of both slow-decomposing Soil Organic Matter (SOM) and fast-decomposing Fresh Organic Matter (FOM). Microorganisms grow through the assimilation of DOM and breath by producing CO2. Upon death, they are transformed back into organic matter.

We recall that for each voxel at the position $p_i$:

- $x_i^1(t)$ is the mass of MB,

- $x_i^2(t)$ is the mass of DOM,

- $x_i^3(t)$ is the mass of SOM,

- $x_i^4(t)$ is the mass of FOM,

- $x_i^5(t)$ is the mass of $CO_2$.

Let $X^j(t) = \left( x_1^j(t), \cdots, x_n^j(t) \right)$ represent the distribution of the $j$ th biological parameter in the pore space. For instance, $X^1(t) = \left( x_1^1(t), \cdots, x_n^1(t) \right)$ is the mass distribution of microorganisms in the pore space. Let $v_i \in V$ be a node of the graph. We model the growth of microorganisms by consuming the available dissolved organic matter according to the Monod equation. Exponential models are employed to represent both death and mass loss due to respiration through mineralized organic matter emission. The total variation of $x_i^1(t)$ is described by the following equation:

$$\frac{dx_i^1(t)}{dt} = -\rho.x_i^1(t) - \mu.x_i^1(t) + \frac{v_{DOM}.x_i^2(t)}{K_{DOM} + x_i^2(t)}.x_i^1(t) \tag{8}$$

Where $\rho$ is the respiration rate, $\mu$ is the mortality rate, $v_{DOM}$ and $K_{DOM}$ are respectively the maximum growth rate of MB and the constant of half saturation of DOM.

In the saturated pore space, dissolved organic matter comes from dead microorganisms, the transformation of soil organic matter, and fresh organic matter. As before, a portion is allocated

from DOM to microorganisms according to the Monod equation. Furthermore, in addition to diffusion processes, the total variation of $x_i^2(t)$ is described by the following equation:

$$\frac{dx_i^2(t)}{dt} = -D_{DOM} \cdot \left[\Delta X^2(t)\right]_i + \beta.\mu.x_i^1(t) - \frac{v_{DOM}.x_i^2(t)}{K_{DOM} + x_i^2(t)}.x_i^1(t) + v_{SOM}.x_i^3(t) + v_{FOM}.x_i^4(t) \quad (9)$$

Where $\Delta = \left(\delta_{i,j}\right)_{1 \le i,j \le n}$ is the Laplacian matrix of the graph $G(V,E)$, $\left[\Delta X^2(t)\right]_i$ is the $i$ th component of $\Delta X^2(t)$, $D_{DOM}$ is the diffusion coefficient of DOM in water, β is the proportion of MB returning to DOM (the other fraction $(1-\beta)$ returns to SOM), $v_{FOM}$ and $v_{SOM}$ the hydrolysis rate of FOM and SOM.

A part of soil organic matter comes from dead microorganisms so the total variation of $x_i^3(t)$ is described by the following equation:

$$\frac{dx_i^3(t)}{dt} = (1-\beta)\mu.x_i^1(t) - v_{SOM}.x_i^3(t) \quad (10)$$

The density evolution of FOM ( $x_i^4(t)$ ) is presented by:

$$\frac{dx_i^4(t)}{dt} = -v_{FOM}.x_i^4(t) \quad (11)$$

The evolution of CO2 ( $x_i^5(t)$ ) is governed by microorganisms breathing, which leads to the following equations:

$$\frac{dx_i^5(t)}{dt} = \rho.x_i^1(t) \quad (12)$$

Our biological model for microbial activity in soil have been validated in [29], and followed the approach of the SOMKO model [30], and MIOR model [31] where microbes are subjected to three physiological processes: survival, maintenance, and growth. In this approach, the CO2 flux is computed as a linear function of biomass.

By gathering all the equations, we have the following system of equations:

$$\begin{cases} dx_i^1(t)/dt = -\rho.x_i^1(t) - \mu.x_i^1(t) \\ \qquad\qquad + \frac{v_{DOM}.x_i^2(t)}{K_{DOM} + x_i^2(t)}.x_i^1(t) \\ \frac{dx_i^2(t)}{dt} = -D_{DOM} \cdot [\Delta X^2(t)]_i + \beta.\mu.x_i^1(t) - \frac{v_{DOM}.x_i^2(t)}{K_{DOM} + x_i^2(t)}.x_i^1(t) \\ \qquad\qquad + v_{SOM}.x_i^3(t) + v_{FOM}.x_i^4(t) \\ \frac{dx_i^3(t)}{dt} = (1-\beta)\mu.x_i^1(t) - v_{SOM}.x_i^3(t) \\ \frac{dx_i^4(t)}{dt} = -v_{FOM}.x_i^4(t) \\ \frac{dx_i^5(t)}{dt} = \rho.x_i^1(t) \end{cases} \quad (13)$$

By removing the diffusion processes of DOM from the equations, and applying an explicit scheme to the first-order derivatives of the equations we obtain the following equations for the transformation processes:

$$
\begin{cases}
x_i^1(t+\delta t) = x_i^1(t) + \dfrac{v_{DOM}.x_i^2(t)}{K_{DOM}+x_i^2(t)}.x_i^1(t).\delta t \\
\qquad\qquad -\left(\rho.x_i^1(t)+\mu.x_i^1(t)\right).\delta t \\
x_i^2(t+\delta t) = x_i^2(t+\delta t) - \dfrac{v_{DOM}.x_i^2(t)}{K_{DOM}+x_i^2(t)}.x_i^1(t).\delta t \\
\qquad\qquad +\left(v_{SOM}.x_i^3(t)+v_{FOM}.x_i^4(t)+\beta.\mu x_i^1(t)\right).\delta t \\
x_i^3(t+\delta t) = x_i^3(t) + (1-\beta)\mu.x_i^1(t) - v_{SOM}.x_i^3(t).\delta t \\
\qquad x_i^4(t+\delta t) = x_i^4(t) - v_{FOM}.x_i^4(t).\delta t \\
\qquad\quad x_i^5(t+\delta t) = x_i^5(t) + \rho.x_i^1(t).\delta t
\end{cases}
\tag{14}
$$

To obtain the voxel states after a time step $\delta t$ of transformation processes, we can apply the equations synchronously or asynchronously. Given that the transformation of the nodes is independent, we have the flexibility to perform transformations sequentially or in parallel on the graph's nodes.

In the second section, we provide a comprehensive explanation of the implementation of all discussed concepts. This includes details on graph construction from the 3D image, the conjugate gradient method, its parallelization, and the implementation of both implicit and explicit schemes. Furthermore, we provide details of implementing transformation processes, whether synchronous or asynchronous.

## 2.4 Implementation details

Generally, simulating transformation-diffusion processes in the complex geometry captured by the 3D image involves transforming the pore space in the image into a graph of connected voxels. Subsequently, each simulation iteration applies transformation processes to the biological attributes of all graph nodes, followed by the solution of the graph diffusion equation to mimic the diffusion of the compound to be diffused in the model (DOM in this study).

Transforming the 3D image into a graph of connected voxels is a straightforward and parallelizable process. It can be accomplished by enumerating the valid voxels in the image (i.e., those corresponding to the pore space). Then, looping over the voxels of the image, when we encounter a valid voxel, we identify from the six neighbouring voxels (Fig 2) the valid ones and construct the adjacency graph.

Let $t \geq 0$ be a specific time, and let $\left(X_i(t)\right)_{i\in N}$ denote the distribution of the nodes at time $t$. The attributes of the nodes, i.e., the distribution of the pore space $\left(X_i(t+\delta t)\right)_{i\in N}$ after a time step $\delta t$ of microbial activity, are obtained by applying the processes in the system of Eq. 14 synchronously (flowchart S3A in S1 File) or asynchronously (flowchart S3B in S1 File). Then, the explicit scheme (Eq. 6) or implicit scheme (Eq. 7) is applied in order to get the distribution after diffusion processes.

In the synchronous procedure for performing microbial transformation, we calculate the contribution of biological variables to uptake, respiration, mortality, and turnover for each node. Subsequently, we update the biological variables according to the logic of the system of Eq. 14.

In contrast, the asynchronous procedure involves sequentially updating the biological variables of the nodes due to each biological process (grow, breath, death …). That means allowing microorganisms to grow by consuming DOM, after which some will die, producing $CO_2$ through respiration, and transforming the dead microbial biomass into DOM and SOM. Detailed flowcharts of the algorithms for both synchronous and asynchronous transformations are described in in flowcharts S3A and S3B in S1 File. When the time steps for both transformation and diffusion are identical, the processes are performed just once per time step. However, if the time steps differ between these processes, we iterate through the process (transformation or diffusion) with the smallest time step to ensure that the dynamics of the entire transformation-diffusion system remain accurate. When using the explicit scheme, the computational process is reduced to straightforward matrix multiplications, that is relatively efficient. On the other hand, implementing the implicit scheme (Eq. 7) requires solving a linear system defined by a large, sparse, symmetric, and positive definite matrix. To handle such computations effectively, we employ the preconditioned conjugate gradient method, where the conditioning matrix $\left(t_{i,j}\right)_{0 \leq i,j \leq n}$, is defined by

$$\begin{cases} t_{i,j} = \dfrac{1}{b_{i,j}} & if\, i = j \quad , \\[2mm] t_{i,j} = 0 & else \end{cases}$$

where $B = \left(b_{i,j}\right)_{0 \leq i,j \leq n}$ is the matrix of the implicit scheme defined in supplementary material S1 File.

The Preconditioned Conjugate Gradient (PCG) method is a variant of the Conjugate Gradient (CG) method used for solving linear systems of equations $Ax = b$, where $A$ is a symmetric positive definite (SPD) matrix. The PCG method incorporates a preconditioner matrix $T$ to improve convergence speed.

The implementation was done using the C language due to its capacity for handling such complex problems: efficient memory management, computational speed required to address some aspects of the simulation like: graph construction from the 3D image, solving the graph diffusion equation through the numerical schemes, and running expensive simulations on the constructed graph.

The C implementation and the image data used in this study can be found at the following links:

https://github.com/mouadklai/VGA_microbial_decomposition

https://www.kaggle.com/datasets/mouadklai/three-image-sandy-loam-soil/data

## 2.5 Diffusional conductance coefficients approximation: theoretical framework

In pore network geometrical modelling, pore space is approximated with a minimal set of maximal geometrical primitives (balls, ellipsoid, cylinders …). Then, a valuated graph is constructed from the set, where the adjacency is the geometrical adjacency between the geometrical primitives. For each pair of geometrically adjacent primitives, we need to determine the conductance, which defines the portion of mass flow to account for when simulating diffusion according to Fick's law.

In this subsection we discuss a possible way to determine approximatively the diffusional conductance coefficients of a pore network geometrical model.

Let $G_{VGA}\left(N_{VGA}, E_{VGA}\right)$ be a voxel graph representation of a 3D image representing a pore space, where $N_{VGA} = \{v_1, \ldots, v_n\}$ represents the valid voxels of the image, $n$ is their number, and $E_{VGA} = \{(i, j) : i, j \in \{1, \ldots, n\} \land v_i \cap v_j \neq \varnothing\}$ encodes the adjacency between them.

Let $G_{PNGM}\left(N_{PNGM}, E_{PNGM}\right)$ be the graph of connected geometrical primitives covering the pore space of the 3D image, where $N_{PNGM} = \{P_1, \ldots, P_q\}$ is the set of geometrical primitives, $q$ is their number, and $E_{PNGM} = \{(i, j) : i, j \in \{1, \ldots, n\} \land P_i \cap P_j \neq \varnothing\}$ encodes the adjacency between them. For all $k \in \{1, \ldots, q\}$ let $V\left(P_k\right) = \{v_i \in N_{VGA} : v_i \in P_k\}$ be the set of voxels contained within the primitive $P_k$. For theoretical explanation, suppose that:

$$k_1, k_2 \in \{1, \ldots, q\}, V\left(P_{k_1}\right) \cap V\left(P_{k_2}\right) = \varnothing$$

$$k \in \{1, \ldots, q\} V\left(P_k\right) = N_{VGA}$$

Let $t \geq 0$ be a time, and $M_{PNGM}(t) = \{m^{PNGM}_1(t), \ldots, m^{PNGM}_q(t)\}$ be a mass distribution of the geometrical primitives $\{P_1, \ldots, P_q\}$. According to Fick's Law of diffusion, the flow of mass between two adjacent primitives $P_i$ and $P_j$, of volume $v_i$ and $v_j$ respectively, at time $t + \delta t$ is given by:

$$F_{i,j} = -Dc.\delta t.\alpha_{i,j} \left( \frac{m_i(t)}{v_i} - \frac{m_j(t)}{v_j} \right) \tag{15}$$

where $D_c$ is the diffusion coefficient, and $\alpha_{i,j}$ is what we call the diffusional conductance between the two primitives. This conductance typically depends on factors such as the contact surface between them, the distance between their centers of mass, the form of the geometrical primitives, and the difference in volume between the two primitives.

Then, the variation of mass distribution $\dfrac{dM_{PNGM}(t)}{dt}$ can be derived in the same way as we have done in Section 1.2, and it is given by:Top of Form

$$\forall i \in \{1, \ldots, q\}, \frac{dm^{PNGM}_i(t)}{dt} = \sum_{j \in \vartheta(i)} -Dc.\alpha_{i,j} \left( \frac{m^{PNGM}_i(t)}{v_i} - \frac{m^{PNGM}_j(t)}{v_j} \right) \tag{16}$$

where $\vartheta(i) = \{j \in \{1, \ldots, q\} : P_i \cap P_j \neq \varnothing\}$ is the indexes of geometrically adjacent primitives to the primitive $P_i$. Numerically, solving the $PNGM diffusion model$, can be done by discretizing time using forward (explicit) or backward (implicit) Euler schemes.

Using the explicit scheme, we get:

$$\forall i \in \{1, \ldots, q\}, \; m^{PNGM}_i(t + \delta t) = m^{PNGM}_i(t) - Dc.\delta t.\sum_{j \in \vartheta(i)} .\alpha_{i,j} \left( \frac{m^{PNGM}_i(t)}{v_i} - \frac{m^{PNGM}_j(t)}{v_j} \right) \tag{17}$$

Then, we have

$$\forall i \in \{1, \ldots, q\}, m^{PNGM k+1}_i = m^{PNGM k}_i - Dc.\delta t.\sum_{j \in \vartheta(i)} .\alpha_{i,j} \left( \frac{m^{PNGM k}_i}{v_i} - \frac{m^{PNGM k}_j}{v_j} \right) \tag{18}$$

Using the implicit scheme, we get:

$$\forall i \in \{1,\ldots,q\},$$

$$m^{PNGM}{}_i(t) = m^{PNGM}{}_i(t+\delta t) + Dc.\delta t. \sum_{j\in\vartheta(i)} .\alpha_{i,j}\left(\frac{m^{PNGM}{}_i(t+\delta t)}{v_i} - \frac{m^{PNGM}{}_j(t+\delta t)}{v_j}\right) \qquad (19)$$

Then, we have

$$\forall i \in \{1,\ldots,q\}, m^{PNGM\,k}{}_i = m^{PNGM\,k+1}{}_i + Dc.\delta t. \sum_{j\in\vartheta(i)} .\alpha_{i,j}\left(\frac{m^{PNGM\,k+1}{}_i}{v_i} - \frac{m^{PNGM\,k+1}{}_j}{v_j}\right) \qquad (20)$$

where $k$ is the iteration number.

The accuracy of the *PNGM diffusion model* depends on the exactitude of calculating the coefficients $\{\alpha_{i,j} : (i,j) \in E_{PNM}\}$.

In [7] and [20] the coefficients $\alpha_{i,j}$ were set to $\alpha\dfrac{S_{i,j}}{d}$, where $S_{i,j}$ represents the contact surface between the primitive $P_i$ and the adjacent primitive $P_j$, $d_{i,j}$ is the distance between the centers of gravity of the spheres $i$ and $j$, and $\alpha$ is a unified coefficient for all adjacent primitives in the network. The value of $\alpha$ was determined to maximize the intercorrelation (Cosinus) $cos_{C_{LBM},C_{PNGM_\alpha}}$ between $C_{LBM}$ and $C_{PNGM_\alpha}$, which are the simulated curves of mass distribution in the z-planes obtained using LBM and PNGM, respectively.

$$cos_{C_{LBM},C_{PNGM_\alpha}} = \frac{C_{LBM}.C_{PNGM_\alpha}}{C_{LBM}.C_{PNGM_\alpha}} = \frac{\sum_{i=1}^{i=512} C_{LBM}(i).C_{PNGM_\alpha}(i)}{\sqrt{\sum_{i=1}^{i=512} C_{LBM}(i)^2}.\sqrt{\sum_{i=1}^{i=512} C_{PNGM_\alpha}(i)^2}} \qquad (21)$$

The maximum value of $cos_{C_{LBM},C_{PNGM_\alpha}}$ was achieved at α = 0.6 in the experiment described in section 3.1, using the dataset presented in this study, that was also employed in [20].

The curve labeled "old PNGM" in Fig 12 represents the results obtained using PNGM. The curve obtained using LBM is not shown, as it closely resembles the one generated using VGA, which is displayed in Fig 12.

In the following, we present a detailed approach to approximate the coefficients using a machine learning approach and data generated using the voxel graph-based approach discussed before.

Let, $M_{VGA}(t) = \{m^{VGA}{}_1(t),\ldots,m^{VGA}{}_n(t)\}$ be the voxel distribution that correspond to the primitive's distribution $M_{PNM}(t)$ at time t, obtained by the voxel description $\{V(P_k) : k \in \{1,\ldots,q\}\}$ of the primitives $\{P_k : k \in \{1,\ldots,q\}\}$.

The evolution of $M_{VGA}(t)$ over time can be calculated using the graph diffusion equation through the use of one of the schemes: the explicit (equation 16) or the implicit (equation 18). The framework for the simulation of $M_{VGA}(t)$ over time was discussed in section 2.2 and validated in section 3.1.

Let $\{(X^p,Y^p) : p \in \{1,\ldots,l\}\}$ be a dataset of couple of distributions: a random distribution $X^p$ of mass in the pore network and the corresponding next distribution $Y^p$ after a time step $\delta t$ of diffusion calculated using the voxel graph-based approach from the corresponding voxel description. Rewriting the scheme 3 for the dataset we have

$$\forall p \in \{1,\ldots,l\}, \forall i \in \{1,\ldots,q\}, y^p{}_i = x^p{}_i - Dc.\delta t. \sum_{j\in\vartheta(i)} .\alpha_{i,j}\left(\frac{x^p{}_i}{v_i} - \frac{x^p{}_i}{v_j}\right) \qquad (22)$$

Finding $\left\{\alpha_{i,j} : (i,j) \in E_{PNGM}\right\}$ is equivalent to finding the set of parameters
$\Theta = \left\{\theta_{i,j} : (i,j) \in E_{PNGM}\right\}$ that minimize for all data $\left\{(X^p, Y^p) : p \in \{1,\dots,l\}\right\}$ one of the following objectives:

$$L_1(\Theta) = \frac{1}{l}\sum_{p=1}^{l}\frac{1}{q}\sum_{i=1}^{q}\left(y^p_{\ i} - x^p_{\ i} + Dc.\delta t.\sum_{j\in\vartheta(i)}.\theta_{i,j}\left(\frac{x^p_{\ i}}{v_i} - \frac{x^p_{\ i}}{v_j}\right)\right)^2 \tag{23}$$

and

$$L_2(\Theta) = \frac{1}{l}\sum_{p=1}^{l}\frac{1}{q}\sum_{i=1}^{q}\left(x^p_{\ i} - y^p_{\ i} - Dc.\delta t.\sum_{j\in\vartheta(i)}.\theta_{i,j}\left(\frac{y^p_{\ i}}{v_i} - \frac{y^p_{\ i}}{v_j}\right)\right)^2 \tag{24}$$

The first objective calculates the L2 loss between the output of the explicit scheme (Eq.18) and the target (next time distribution) obtained using VGA.

The second objective minimizes the error between the input and second member of the implicit scheme (Eq. 20) applied to the target (next time distribution) obtained using VGA.

Stochastic Gradient Descent (SGD) is an efficient optimization algorithm widely used in machine learning to minimize loss functions. Unlike standard Gradient Descent, which computes gradients using the entire dataset, SGD updates parameters more frequently by using a single data point or a small batch, making it faster but introducing noise.

The gradient for an objective function involving a dataset $\left\{(X^p, Y^p) : p \in \{1,\dots,l\}\right\}$ (with l representing batch size) is computed as follows

$$\frac{d}{d\theta_{i,j}}L_1(\Theta) = \frac{1}{l}\sum_{l}^{p=1}\left[\frac{2}{q}Dc.\delta t\left(\frac{x^p_{\ i}}{v_i} - \frac{x^p_{\ j}}{v_j}\right)\times\left(y^p_{\ i} - x^p_{\ i} + Dc.\delta t\sum_{f\in\vartheta(i)}\theta_{i,f}.\left(\frac{x^p_{\ i}}{v_i} - \frac{x^p_{\ f}}{v_f}\right)\right)\right] \tag{25}$$

And the gradient of the second objective is calculated using the following:

$$\frac{d}{d\theta_{i,j}}L_2(\Theta) = \frac{1}{l}\sum_{l}^{p=1}\left[\frac{2}{q}Dc.\delta t\left(\frac{y^p_{\ i}}{v_i} - \frac{y^p_{\ j}}{v_j}\right)\times\left(x^p_{\ i} - y^p_{\ i} - Dc.\delta t\sum_{f\in\vartheta(i)}\theta_{i,f}.\left(\frac{y^p_{\ i}}{v_i} - \frac{y^p_{\ f}}{v_f}\right)\right)\right] \tag{26}$$

In each training step, we calculate the set $\left\{\frac{d}{d\theta_{i,j}}L(\Theta) : (i,j) \in E_{PNM}\right\}$, and we update the parameters using the following stochastic gradient descent policy:

$$\forall (i,j) \in E_{PNGM}, \theta_{i,j} = \theta_{i,j} - lr\frac{\dfrac{d}{d\theta_{i,j}}L(\Theta)}{\left|\dfrac{d}{d\theta_{i,j}}L(\Theta)\right|}. \tag{27}$$

The use of the normalized gradient $\dfrac{\frac{d}{d\theta_{i,j}}L(\Theta)}{\left|\frac{d}{d\theta_{i,j}}L(\Theta)\right|}$ instead of the gradient $\frac{d}{d\theta_{i,j}}L(\Theta)$ addresses

issues related to extremely small gradient magnitudes. This normalization helps in maintaining numerical stability, ensuring consistent and meaningful updates to the parameters, and potentially improving the convergence rate of the optimization process.

Using the attained value of the objective function $L(\Theta)$ at each training iteration, we decide whether to stop training, continue, or adjust the learning rate ($lr$) and restart training.

After the training using stochastic gradient descent, we extract the learned parameters that will approximate the coefficient $\left\{\alpha_{i,j} : (i,j) \in E_{PNGM}\right\}$.

Note that the estimated diffusional conductance coefficients are theoretically applicable across all diffusion scenarios, regardless of the chosen diffusion coefficient or time step used in training, as they remain independent of these parameters.

In results and discussion section we test the discussed theoretical framework in predicting the diffusional conductance coefficient for the ball network model discussed before.

## 2.6 Dataset description

To compare the performance of the implemented numerical schemes, we used 3D microcomputed tomography (µCT) image data obtained from an experimental sandy loam soil collected at the Bullion Field site, located at the James Hutton Institute in Invergowrie, Scotland. The dataset was previously used in studies such as [4,7]20].

**2.6.1 Soil and imaging characteristics.** The Bullion Field soil sample is classified as sandy loam, comprising 71% sand, 19% silt, and 10% clay by mass. This soil texture is ideal for investigating microbial activity due to its heterogeneous pore structure. The µCT scan was performed at a pixel resolution of 24 µm, which is sufficiently high to capture the range of pore sizes associated with microbial habitats and nutrient diffusion. A cubic sub-volume of $512^3$ voxels was extracted from the 3D stack, corresponding to a physical volume of approximately 1.855 cm³.

**2.6.2 Image segmentation and pore structure analysis.** To differentiate the solid and pore phases within the soil, we applied the indicator kriging method [32], yielding a segmented 3D binary image. The segmentation reveals visible 17% and 8% porosities for soil bulk densities of 1.2 g cm$^{-3}$ and 1.6 g cm$^{-3}$, respectively. The segmented image was analyzed to extract pore network characteristics, particularly focusing on maximal inscribed spheres within the pore space, following the methodology outlined in [21]. This approach ensures the extracted pore network accurately represents the critical pathways for microbial movement and nutrient diffusion.

**2.6.3 Boundary conditions and simulation setup.** In the simulations, the soil sample was treated as a closed environment, which is a critical consideration for the boundary conditions applied. This assumption was made to reflect the controlled conditions typically found in experimental settings, allowing for more precise comparisons of microbial dynamics and diffusion processes within the soil structure.

Figs 4 and 5 in the manuscript illustrate representative cross-sections and 3D views of the binary image, highlighting the heterogeneous pore structure. For pore network geometrical modeling, Fig 6 depicts a view of the ball network wherein we specifically choose the balls whose centers reside within the region bounded by $\left[[50,100],[50,100],[150,200]\right]$.

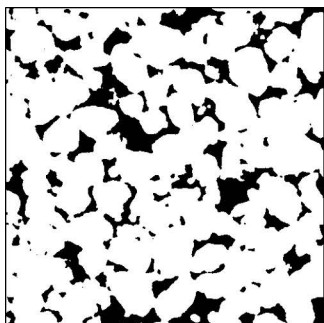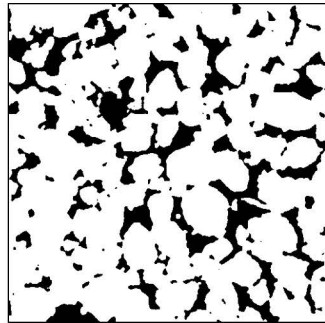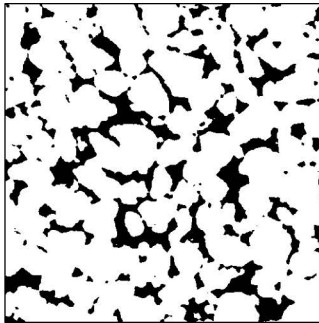

**Fig 4. Random cross-sections of the 3D binary image depicting pore space (black) and solid matrix (white).**
These slices are taken from random depths within the 3D structure, highlighting the spatial variability and heterogeneity of the pore network. The image illustrates the complexity of the porous medium involved in simulating microbial activity in soil.

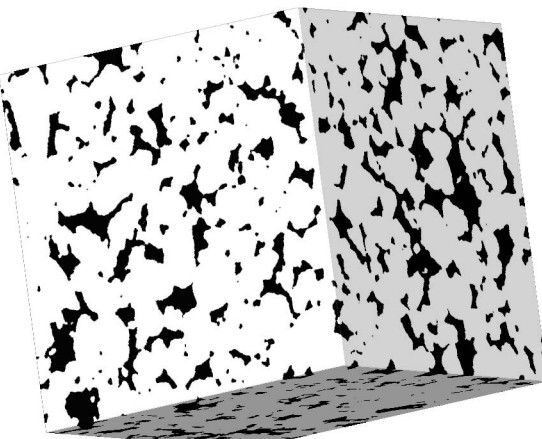

**Fig 5. Three-dimensional visualization of the binary image, illustrating the spatial distribution and morphology of two distinct phases within a cubic volume. The black regions represent the void (pore space), while the white regions represent the solid phase.**

## 3. Results and discussion

In this section, we compare simulations using the presented approach that we call the Voxel Graph-based Approach (VGA), with LBM-based simulation and pore network-based simulations. For LBM-based simulation, we use the approach outlined in [22], where diffusion is computed from the lattice-Boltzmann equation, and transformation is calculated using the synchronous scheme.

For the pore network geometrical model, we use the generic tool discussed in [29]. This tool models the complex shape of the pore space using a graph of connected balls, and formulate diffusion using the first Fick's law. The mass transport between the balls was calibrated to fit the maximum intercorrelation between simulated mass distributions after a simulation time using the ball approach and LBM approach [20].

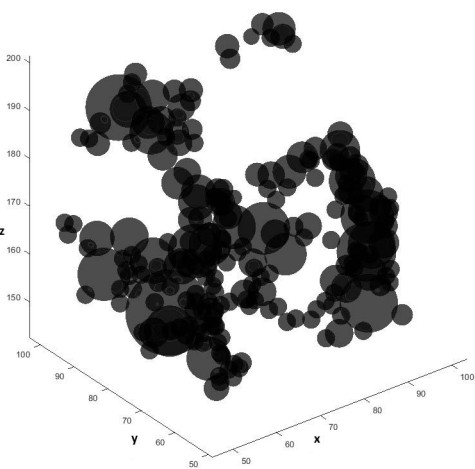

**Fig 6. 3D plot of the spheres set approximating the pore space of the 3D binary image with centers located within the region bounded by [50,100] on the x-axis, [50,100] on the y-axis, and [150,200] on the z-axis.**

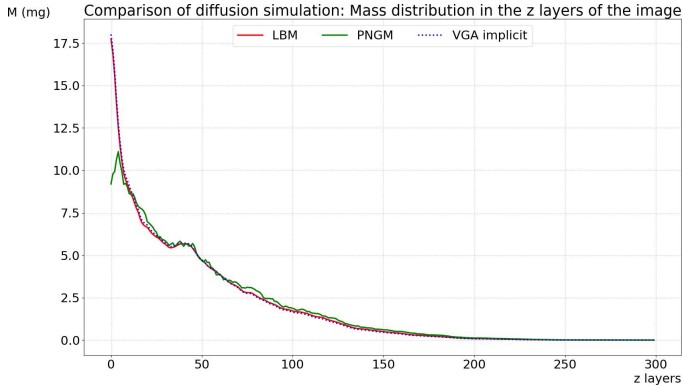

**Fig 7. Comparison of diffusion simulations: implicit scheme of the GDE with a 30-second time step (blue line), implicit scheme of the PNGM with α = 0.6 and a 15-second time step (green line), and LBM with a 0.43-second time step (red line).**

## 3.1 Comparison of diffusion simulations: VGA, LBM and PNGM-based simulations

To compare diffusion simulation using the graph diffusion equation with simulations using the Lattice Boltzmann method and the pore network model, we draw upon a prior experience detailed in [20], which was conducted to compare diffusion using a pore network model (ball network model) and LBM, and we outline our approach as follows:

Uniformly, we distributed $M_0 = 592.7593 mg$ of DOM on the first two z layers ($z = 1$ and $z = 2$) of the $512 \times 512 \times 512$ image. In our approach and the LBM, we selected all the voxels of the two layers ($z = 0$ and $z = 1$) corresponding to the pore space and distributed $M_0$ uniformly among the selected voxels. For the ball network model, we selected the balls intersecting the two layers and distributed the mass $M_0$ homogeneously across them (i.e., applying the same concentration to all the balls). The experiment aims to compare mass profiles in

each layer of the image after some time of $1.76 hours$ . We ran the simulations using the three methods and obtained the final distributions, from which we calculated the mass in each layer of the image. Subsequently, we plot these results together for comparison. The basic principle of pore network geometrical modelling (PNGM) is to globally transport mass from one ball to the connected balls following Fick's laws. Therefore, we need to calibrate the diffusional conductance. We employed a general coefficient $\alpha = 0.6$ , which we multiplied by the contact surface area between all pair of geometrically connected balls, calibrated from LBM [20].

Details on how the coefficient α is determined are provided in Section 2.4.

Fig 7 illustrates a global comparison of the overall accuracy of each method. PNGM was simulated using the implicit scheme with a time step of 15 seconds, while GDE was solved using the implicit scheme with a time step of 30 seconds. In contrast, the time step for LBM is 0.43 seconds, calculated based on the 3D image resolution. The results obtained with the implicit scheme of the GDE are very close to those of LBM and much more accurate than those of the implicit scheme of PNGM. The use of voxels to discretize the space is similar to LBM; however, our approach models diffusion as mass transport according to Fick's law, whereas LBM uses particle-based collision and propagation processes. Although both methods are conceptually similar, our approach involves less calculations.

In contrast, PNGM uses geometrical primitives that are larger than a single voxel and vary in size. This approach makes modeling mass transport in the pore space using Fick's law less precise.

The explicit scheme for solving the graph diffusion equation requires the use of small-time steps to maintain stability and accuracy in the numerical solution. This necessity arises because each time step needs to be small enough to ensure that the diffusion process is accurately captured and does not lead to instability or inaccuracies in the results. As a result, the explicit scheme involves frequent updates and calculations, which can become computationally expensive, especially for large-scale problems or long simulation times.

In contrast, the implicit scheme offers a more computationally efficient alternative by allowing the use of extended time steps. This approach improves computational efficiency because it reduces the number of required updates and calculations, as larger time steps cover a greater extent of the diffusion process in each iteration. Although implicit methods are typically more complex to implement, they often provide better stability and accuracy over longer time periods.

Fig 8 illustrates the comparison between these two schemes. In the explicit scheme, a time step of 0.1 seconds is used, resulting in frequent updates and higher computational costs. On the other

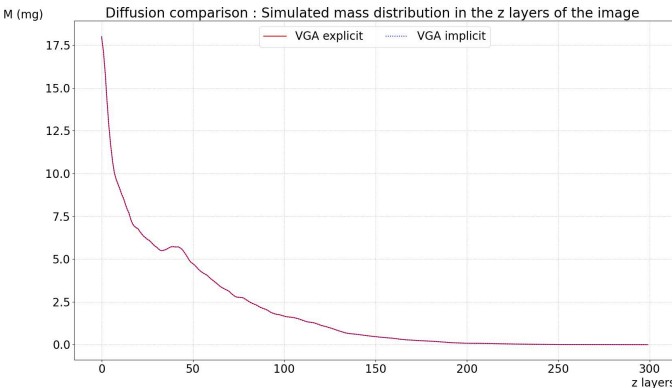

**Fig 8. Comparison of diffusion simulation using the implicit and explicit schemes of the graph diffusion equation: the implicit scheme time step was set to 30 seconds, while the explicit scheme time step was set to 0.1 seconds.**

hand, the implicit scheme employs a relatively extended time step of 30 seconds, demonstrating a reduction in computational effort while still maintaining accuracy in the simulation results.

## 3.2 Comparison of microbial decomposition simulations: VGA, LBM, and PNGM-based simulations

In this section, we validate the efficiency of our method for simulating microbial decomposition of organic matter by comparing its results with those obtained using the LBM-based approach and the PNGM-based method. We conducted numerical simulations of microbial decomposition using the LBM-based approach, the Voxel Graph-based Approach (VGA), and the ball network-based approach on the same dataset previously discussed.

The simulation procedure is as follows: Initially, we evenly distributed $M_2^0 = 289.5 \mu g$ of DOM and introduced $1000$ bacterial spots (clusters) into the pore space at random locations. This patchiness of bacterial distribution, as a result of cell growth mechanisms and environmental constraints, has been investigated in [33]. The clusters collectively represent $5.2 \times 10^7$ bacteria cells, which correspond to $M_1^0 = 5.2 \times 10^7 \times 5.41 \times 10^{-8} \mu g = 2.8132 \mu g$ of carbon.

For the PNGM-based approach, we distributed the mass $M_2^0$ homogeneously, (i.e., the same concentration in each ball), and we put bacterial mass into the balls corresponding to the random location of the spots.

The total mass in the pore space initially is as follows: $\left( M_1^0, M_2^0, M_3^0, M_4^0, M_5^0 \right)$, where $M_3^0 = M_4^0 = M_5^0 = 0$, corresponding to $99.0376\%$ of dissolved organic matter and $0.962\%$ of living microorganisms.

The biological parameters employed in this study were adopted from [29] for Arthrobacter sp. 9R that was calibrated from real experimental data. These parameters are as follows:

- $\rho = 0.2 day^{-1}$ is the relative respiration rate,

- $\mu = 0.5 day^{-1}$ is the relative mortality rate,

- $\beta = 0.55$ is the proportion of microbial mass (MB) returning to DOM (the remaining portion is for MB returning to SOM),

- $v_{SOM} = 0.01 day^{-1}$ and $v_{FOM} = 0.3 day^{-1}$ are the relative decomposition rates for SOM and FOM respectively,

- $v_{DOM} = 9.6 day^{-1}$ and $K_{DOM} = 0.001 gC.g^{-1}$ are the maximum relative growth rate of MB, and the half saturation constant of DOM respectively.

- $D_{DOM} = 100950 voxel^2.day^{-1}$ is the diffusion coefficient of DOM in water. The value used corresponds to the diffusion coefficient of an organic molecule in water [34].

Parameterization of our model has been performed on experimental measurements of growth and respiration curves of Arthrobacter sp cultivated in optimized substrate conditions [29]. This implies that the calibrated respiration rate value lumps together maintenance (renewal of biomass synthesis) and growth (new biomass synthesis) respiration.

We conducted simulations using the three different methods over a 5-day period. For the LBM-based method a time step of 0.43 seconds is used for diffusion and the synchronous transformation processes with the same time step (Fig 9). For the ball network model, we employed the calibrated implicit diffusion scheme from [20] and the asynchronous algorithm for transformation simulation with a 5-second time step.

For our approach, we conducted tests using different time steps to understand their impact on simulation results (see Appendix S2 in S1 File for more details). Table S2T1 provides details on these

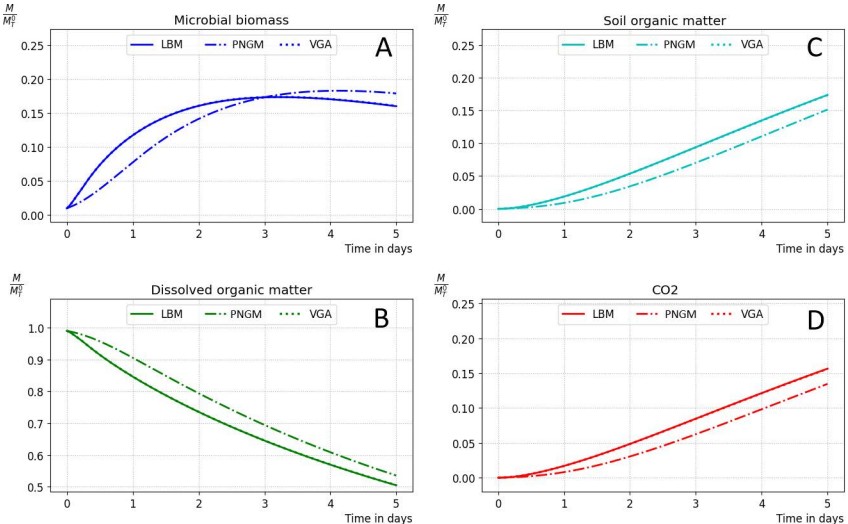

**Fig 9. LBM-based approach using synchronous transformation with a time step of 0.43s, PNGM-based method using asynchronous transformation with a 5s time step, VGA using explicit scheme with a 0.1s time step and asynchronous transformation with a 0.43s time step.**

time steps along with the elapsed time for each simulation. On a standard PC with an AMD Ryzen 7 PRO 6850H processor and 32.0 GB of RAM, the computation times are approximately as follows:

- LBM-based simulation: 3 weeks

- VGA simulation using an explicit scheme (test 3): 7.3 days

- VGA simulation using an implicit scheme (test 1): 5.2 days (refer to Table S2T1 in Appendix S2 in S1 File for more details)

- Pore Network Model (PNGM)-based method: 45 minutes

The optimal intercorrelation was achieved in test 3 (see Appendix S2 in S1 File for more details), which is expected because it used the same time step as the LBM-based simulation, specifically 0.43 seconds for transformation. The decision to use a time step of 0.1 seconds for the explicit scheme was based on two factors: first, the accuracy of the explicit scheme, as it does not require approximation; second, 0.1 seconds represents the largest possible time step for the explicit scheme (at least for this dataset and experimental setup), otherwise we get negative values.

In contrast, Test 4 demonstrates that an implicit scheme with the same time step as that used for the pore network model (5s) produces results that are significantly divergent from both LBM and PNGM simulations (Fig S2C in S1 File).

The voxel graph-based approach presents a practical alternative to LBM for simulation of microbial decomposition of organic matter in soil, as it yields comparable results at lower computational costs. The results obtained indicate that VGA outperforms the other two methods by striking a balance between accuracy and computational time. This superiority can be attributed to three main factors: firstly, the representation of the pore space using a graph of connected voxels, employing only a 6-connectivity policy, results in a more accurate representation than, the pore network models and a lighter representation compared to the lattices used in LBM (D3Q19 in this case). Secondly, simulating mass transport between connected voxels using Fick's law is computationally less expensive than simulating discrete collisions and particle propagation within a lattice

grid. Finally, the implicit scheme of the graph diffusion equation, which allows the use of larger time steps than those used in LBM, is a key factor in reducing the number of iterations.

The PNGM-based approaches demonstrate significant cost-effectiveness compared to voxel-based approaches such as the LBM-based approach or the discussed voxel graph-based approach. Even time steps of 30 seconds still yield good results (refer to [20] for more details).

In the next section, a detailed theoretical approach will be provided for calculating the diffusional conductance coefficients for PNGM models using the voxel graph-based approach, in order to improve accuracy.

### 3.3. Improving PNGM simulations

In the context of the ball network detailed in Section 2.4, we initialize the parameters denoted as $\Theta = \left\{ \theta_{i,j} : (i,j) \in E_{PNGM} \right\}$ by

$$\forall (i,j) \in E_{PNGM}, \theta_{i,j} = \frac{S_{i,j}}{d_{i,j}}$$

Where $S_{i,j}$ represents the contact surface and $d_{i,j}$ signifies the distance between the two connected primitives $i$ and $j$. Subsequently, a synthetic 3D pore space image is generated to represent the balls, followed by the construction of the voxel graph as outlined in section 2. Thirty scenarios of Distribution of Organic Matter (DOM) within the pore space voxels are generated as follows: for each scenario, a random mass $M_0$ is generated and distributed randomly among the voxels. The diffusion process is simulated using the explicit scheme of the voxel graph diffusion equation with a time step of 0.1 seconds and a diffusion coefficient $D_c = 100950 voxel^2 . j^{-1}$. The mass distribution in the voxels is recorded every 10 seconds during the simulation.

Subsequently, all resulting voxel distributions are mapped to the corresponding ball distributions. These ball distributions are used to minimize the implicit objective $L_2$ (Eq. 24).

A total of 3600 data points is obtained, each consisting of a mass distribution in the ball network and the corresponding mass distribution after 10 seconds of diffusion simulated using the accurate VGA-based simulation.

We run 1000 epochs of stochastic gradient descent for training. Each epoch involves the selection of four random data points from the obtained dataset, then calculating the

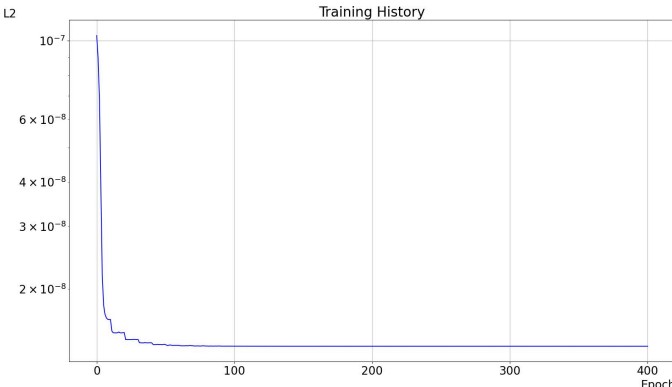

**Fig 10. Training History: The x-axis represents the number of epochs, and the y-axis represents the L2 error for the four selected data points.**

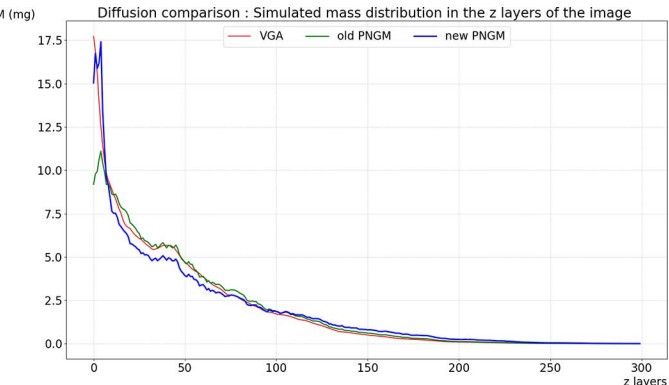

**Fig 11. Diffusion simulation comparison: VGA-based simulation (red curve), PNGM-based simulation calibrated by LBM-based simulation (green curve), and PNGM-based simulation with the approximated diffusional conductance coefficients (blue curve).**

normalized gradient and updating the learning parameters accordingly. Training starts with

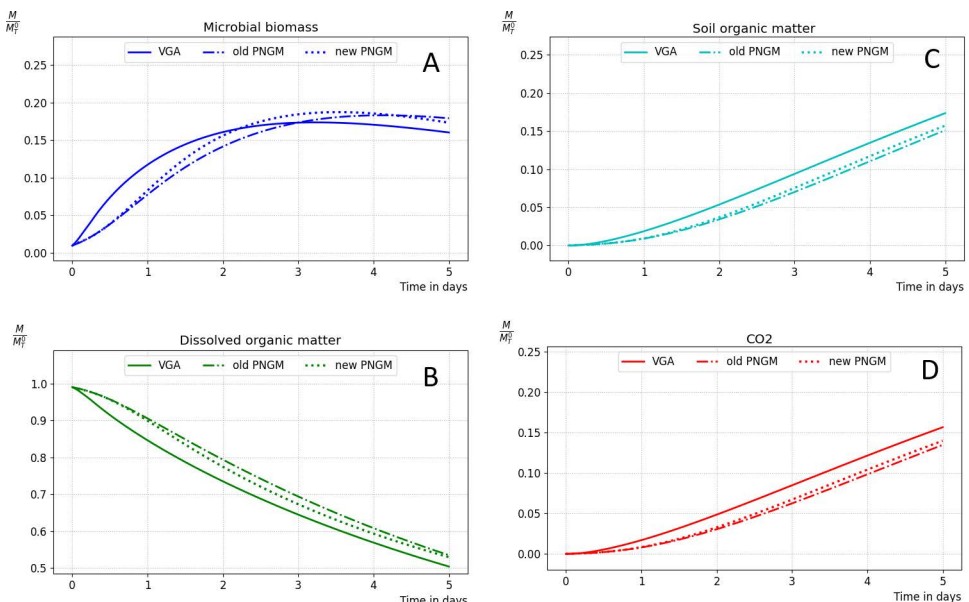

**Fig 12. Microbial decomposition simulation: VGA-based simulation (solid lines), PNGM-based simulation calibrated by LBM-based simulation (dash-dotted lines), and PNGM-based simulation with the approximated diffusional conductance coefficients (dotted lines).**

a learning rate of 0.1, which is then halved every 10 epochs. During training, we calculate the $L_2$ loss applied to four chosen data points to track minimization. The history of the $L_2$ values is plotted in Fig 10. The loss decreases significantly indicating rapid learning and convergence within the first 10 epochs. Also, the training shows great stability and convergence to the solution, with a minimal $L_2$ value of $1.37766510 \times 10^{-8}$.

**Table 1. Pros and Cons of discussed numerical approaches for simulating microbial decomposition of organic matter in soil.**

| Approach | Pros | Cons |
|---|---|---|
| LBM | • Excellent for simulating diffusion and advection in porous media. | • Memory-intensive for 3D CT data.<br>• Time-consuming due to small time steps.<br>• Complex to implement. |
| PDE solvers | • Direct simulation of reaction-diffusion systems.<br>• Mesh refinement improves accuracy in critical regions. | • Expensive and time-consuming meshing.<br>• Difficult to define boundary conditions for irregular media.<br>• Complex to implement. |
| PNGM | • Simplifies porous media by representing pore spaces as networks of connected pores.<br>• Efficient for large-scale simulations.<br>• Less computationally intensive. | • Eliminates small-scale geometrical features.<br>• Challenging to extract and validate pore networks geometrical models from 3D CT images.<br>• Requires good calibration. |
| VGA | • Exact representation of pore space from 3D CT images.<br>• Simple, less expensive diffusion modelling through Fick's law.<br>• Tolerate larger time steps (Implicit scheme) which enable long-term simulations.<br>• Easy to implement. | • Does not account for advection.<br>• Limited to two-phase systems (liquid-solid). |

After epoch 355, the learning process stabilizes, and the stochastic gradient descent algorithm converges to the set $\Theta = \left\{ \theta_{i,j} : (i,j) \in E_{PNGM} \right\}$ that minimize the application $\Theta \mapsto L_2(\Theta)$ for the data provided. The obtained coefficients replace the coefficients $\left\{ \alpha_{i,j} : (i,j) \in E_{PNGM} \right\}$ in one of the schemes (either Eq. 17 or Eq. 19) to simulate diffusion within the ball network, which serves as an approximation of the pore space.

Since the number of spheres is considerably smaller than the number of voxels in the pore space, simulations using spheres require less computational effort than those using voxels. However, accurate diffusion simulations within the sphere network rely on precisely determining the diffusional conductance coefficients [7,20,24–26].

Figs 11 and 12 compare simulation results using two different sets of diffusional coefficients: the previous coefficients from [20] and the new coefficients derived using stochastic gradient descent based on data generated from the voxel approach. These results are then compared with the results of the accurate VGA.

Fig 11 presents a comparison of simulated mass after 1.73 hours of diffusion, using the same scenario described in Section 3.1, with both VGA-based and PNGM-based simulations. The figure includes results from VGA using explicit scheme with a 0.1s time step a PNGM model calibrated by LBM simulation (referred to as the old PNGM) and from a PNGM model with diffusional conductance coefficients obtained after 1000 epochs of training (referred to as the new PNGM) using explicit scheme with a 0.1s time step.

The new PNGM shows an improvement over the old PNGM, demonstrating enhanced accuracy in the simulation results. The difference between the VGA and PNGM curves near the origin is due to the limited number of spheres intersecting the first plane, causing the initial DOM mass to be more concentrated in the second plane. Specifically, in the VGA approach, for each voxel containing organic matter, the nearest sphere in the PNGM is identified, and the corresponding organic matter is placed accordingly.

We conducted microbial decomposition simulations under the same scenario as in Section 3.2, employing the old PNGM-based simulation calibrated using LBM in previous work [20], and the improved PNGM-based simulation using the newly obtained diffusional

conductance coefficients (Table 1). These simulations were done using the implicit scheme with a 10s-time step and the asynchronous transformation using a 30s-time step. The results of these simulations are compared with those of the VGA-based simulation using the explicit scheme with a 0.1s-time step and the asynchronous scheme of transformation with a 0.43s-time step.

Fig 12 compares simulated microbial decomposition, using both VGA and PNGM. The figure includes results from PNGM calibrated using LBM simulations as described before (labeled "old PNGM"), PNGM with diffusional conductance coefficients optimized through training (labeled "new PNGM"), and VGA.

While the "new PNGM" model maintains the structural characteristics of the "old PNGM" model, its curves are consistently closer to the VGA-based simulation results across all components of microbial decomposition. This indicates that the optimization of diffusional conductance coefficients improved the model's performance.

The following table summarizes the pros and cons of all the simulations approaches discussed in this paper.

## 4. Conclusion

This study introduces the Voxel Graph-based Approach (VGA) to simulate microbial activity using the set of voxels of pore spaces obtained using 3D computed tomography images of soil. Traditional methods such as the Lattice Boltzmann Method (LBM) or PDE solvers are computationally demanding when dealing with complex 3D structures or large image data. Pore Network Geometrical Models (PNGM), while more efficient, often require calibration and suffer from reduced accuracy due to oversimplified geometrical representations. VGA addresses these issues by considering an exact representation of the pore space using a graph of connected voxels directly from 3D micro-CT images, allowing accurate modeling of transport processes based on Fick's law. When coupling the transport processes with biodegradation model, our approach simulates microbial decomposition of organic matter in saturated porous media accurately and faster than traditional methods.

Our comparative analysis demonstrates that VGA achieves comparable accuracy to LBM while reducing computation time by a quarter. Although VGA is slower than PNGM, it provides superior accuracy and requires no calibration.

PNGM-based simulations can yield results close to those of VGA or LBM when a robust calibration method is applied. Notably, we showed that VGA improves PNGM-based simulations by refining diffusional conductance coefficients through stochastic gradient descent, which can be used to improves the precision of pore network geometrical modeling in large-scale applications.

The VGA approach offers several key advantages:

- **Refining Diffusional Conductance Coefficients:** VGA generates high-accuracy data with minimal computational cost, which can be used to approximate diffusional conductance coefficients in PNGM-based simulations.

- **Incorporating Pore Space Deformation:** Unlike PNGM, which relies on rigid geometric primitives, VGA represents pore space as a graph of connected voxels. This representation allows for dynamic changes, such as voxel movement or removal, enabling the modeling of pore space deformation.

- **Upscaling:** By integrating VGA with graph neural networks, we can train models to predict long-term microbial dynamics from 3D images of larger samples, addressing upscaling challenges effectively.

- **Generalizing:** VGA can simulate a wide range of diffusion and transformation processes in two-phase porous media or fractured environments.

However, VGA also has limitations:

- **Spatial Discretization:** The accuracy of the model is influenced by the spatial discretization, determined by the voxel resolution. In our case, the voxel size of 24 μm may limit the resolution of small-scale features.

- **Two-Phase System:** The current implementation is restricted to a two-phase system, which may not fully capture the complexity of multi-phase systems found in real-world scenarios. However, it is straightforward to extend the framework by incorporating additional phases using a heterogeneous graph approach, which we plan to explore in the future.

- **Advective Transport:** The VGA framework does not account for advective transport processes, which can be significant in many environmental and biological systems.

Despite the limitations, VGA provides a valuable tool for simulating microbial decomposition of organic matter in complex 3D soil structures. Compared to recent methods by Pot et al. (2022), Monga et al. (2022), and Zech et al. (2022) that is restricted to 2D applications, the voxel graph-based approach discussed in this paper is more efficient for simulating microbial decomposition of organic matter in 3D porous media obtained from computed tomography images of soil.

## Supportimg information

**S1 File.  Appendices S1, S2 and S3.**
(DOCX)

## Acknowledgment

This research was supported by **I-MAROC project** (**APRD program, CNRST/OCP**) and **MICROLARGE** project (**ANR-23-CE01-0018-01**, French national agency ANR). We thank Pr. Patricia GARNIER and Pr. Philippe BAVEYE for fruitful discussions. The scholarship grant of Mouad KLAI was financed by the international doctoral program scholarship (PDI, Sorbonne Université/ IRD).

## Author contributions

**Conceptualization:** Olivier Monga.

**Data curation:** Mouad Klai.

**Formal analysis:** Mouad Klai, Olivier Monga.

**Investigation:** Olivier Monga.

**Methodology:** Mouad Klai, Valérie Pot.

**Project administration:** Olivier Monga.

**Supervision:** Olivier Monga, Mohamed Soufiane Jouini, Valérie Pot.

**Validation:** Mouad Klai, Valérie Pot.

**Writing – original draft:** Mouad Klai.

**Writing – review & editing:** Olivier Monga, Mohamed Soufiane Jouini, Valérie Pot.

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
