## [Decision Letter · Decision Letter 0]

18 Jul 2024

PONE-D-24-24748A Voxel-based Approach for Simulating Microbial Decomposition in Soil: Comparison with LBM and Improvement of Morphological ModelsPLOS ONE

Dear Dr. Jouini, Thank you for submitting your manuscript to PLOS ONE. After careful consideration, we feel that it has merit but does not fully meet PLOS ONE’s publication criteria as it currently stands. Therefore, we invite you to submit a revised version of the manuscript that addresses the points raised during the review process.

We look forward to receiving your revised manuscript.

Kind regards,

Eiyad Ahmad Moh’d Abu-Nada

Academic Editor

PLOS ONE

Journal Requirements:

2. Please note that PLOS ONE has specific guidelines on code sharing for submissions in which author-generated code underpins the findings in the manuscript. In these cases, all author-generated code must be made available without restrictions upon publication of the work. 

Please review our guidelines at https://journals.plos.org/plosone/s/materials-and-software-sharing#loc-sharing-code and ensure that your code is shared in a way that follows best practice and facilitates reproducibility and reuse.

4. Please ensure that you refer to Figures 9 to 11 in your text as, if accepted, production will need this reference to link the reader to the figure.

**Additional Editor Comments:**

The authors must provide a detailed response for each point raised by the three reviewers.

Reviewers' comments:

Reviewer's Responses to Questions

**Comments to the Author**

1. Is the manuscript technically sound, and do the data support the conclusions?

Reviewer #1: Yes

Reviewer #2: Yes

Reviewer #3: Partly

2. Has the statistical analysis been performed appropriately and rigorously? 

Reviewer #1: I Don't Know

Reviewer #2: N/A

Reviewer #3: N/A

3. Have the authors made all data underlying the findings in their manuscript fully available?

Reviewer #1: Yes

Reviewer #2: No

Reviewer #3: No

4. Is the manuscript presented in an intelligible fashion and written in standard English?

Reviewer #1: Yes

Reviewer #2: No

Reviewer #3: Yes

5. Review Comments to the Author

Reviewer #1: This study presents a novel computational method for simulating microbial decomposition in soil using 3D micro-CT images. It uses a valuated graph of connected voxels to model complex diffusion and transformation processes. The method includes parallelization strategies and evaluates various numerical techniques. Validation against LBioS (lattice-Boltzmann method) and Mosaic (Pore Network Geometrical Modelling) shows it matches LBM-based simulation accuracy with a quarter of the computing time and exceeds Mosaic in accuracy without calibration. Additionally, it enhances PNGM-based simulations by approximating diffusional conductance coefficients using stochastic gradient descent. Based on the initial assessment of this manuscript, the current decision is a major revision to reconsider the publication with Plos One.

Abstract

1. Line 18-19: It would be good to point out the research gap before running into aims at the beginning of your abstract (be concise, please; after all, it is just an abstract).

2. Line 25: Could you please define LbioS in your abstract as it first appears in your content?

3. Line 28: one-fourth -> a quarter

4. Line 33: It would be better if you could restate/highlight your engineering contribution from the novel method you developed in this study (please be concise; after all, it is just an abstract).

Introduction

5. Line 51: These efforts aim to … ? or These studies aim to … ? Can efforts aim to do something?

6. Line 68: You have not accomplished your literature review by identifying a research gap in high clarity. You have not compared and contrasted the pros and cons from the historical track of previous studies (the initial, later progress, and state of the art by now; what has been done already and what else has not been achieved yet?) in order to identify those research gaps before you propose your research objectives, followed by methods from Line 69. Your review of published works should solidly support your research purposes before heading into it. There is a significant gap or logical disconnection between Lines 68 and 69.

7. Lines 82-87: You have structurally illustrated your paper in the second last paragraph without mentioning the specific sections/chapters. You don’t need to redo it with an independent paragraph that looks slightly redundant.

8. At the end of your Introduction and right after your research proposal, you should re-highlight your research significance and engineering contributions to dazzle the reviewers and readers, who have short memories after reading a series of lengthy content, because not all of them can easily recall all the content you provided in the background and literature review in the Introduction. Note that it must be concise and informative.

Material and methods

9. Lines 91-92: You can just write, “In this study, we only considered the condition that soil is fully saturated, i.e., soil pore space is filled with water, in exception that soil under the unsaturated condition, i.e., soil void space is partially filled with soil moisture.” An aquatic environment could represent many other meanings, not only in topsoil, vadose zone, and groundwater under the phreatic line (i.e., groundwater table) but also in other water bodies, e.g., nearshore seawater, surface flow, etc.

10. For a numerical contribution like this, I recommend you relocate your pseudo code (algorithm) into an appendix and leave a Matlab or Python code in your supplementary file so many researchers interested in this code will further work on it. It would be much better if you could draw a flow chart for your pseudo-code as a figure in the main text rather than in the appendix and supplementary material.

11. Since it was not observable in your figures in terms of a few very important key points for a numerical simulation, e.g., boundary conditions, initial conditions, etc., you should better visualize those in your simulation demo figures and also provide other information regarding convergence and numerical stability issues (e.g., size of discretization, temporal step in your implicit and explicit schemes, simulating scale compared to single particles or the representativeness of REV, etc.) in the numerical method you just developed. Those will help you defend your work from other peer reviewers who are experts in developing numerical schemes.

Results and discussion

12. As you used image analysis on CT scans, have you provided adequate technical details on CT scanning and image processing, e.g., voxel-length calibration, ratio, measuring precision, etc.?

13. Line 516: You claimed you adopted the D3Q9 scheme for LBM simulation. So far, to the best of my knowledge, it should have D2Q9, D3Q15, and D3Q27. Could you please visualize your D3Q9 cell in 3D space, as you demonstrated in Figure 5? Please don’t forget to update all figure numbers in both their captions and content.

14. As for many diffusion coefficients or diffusivities you adopted in your numerical simulations, I recommend you defend your selection of those by comparing them against published works, i.e., necessary citations of new references should be added in. This question could also be raised to select other parameters of the properties of fluid and porous media in your work. It will convince experts who have more experience in this area.

15. Lines 734-754: The pros and cons could be better summarized in a table for comprehensive comparison and contrast. You can also do so to compare different numerical methods. It will shorten the content you must deliver to readers and make the paper more concise and informative, i.e., straightforward to read and check all essentials in just a few tables.

Conclusions

16. Please reconstruct your concluding section as follows:

Paragraph 1: Concise summary of your research gaps identified through literature review and research objectives; Briefing your methodology with some very critical settings;

Paragraph 2: Listing your conclusions or research findings point-by-point;

Paragraph 3: Mention the limitations and reflection of your work; Highlight the significance of your research and engineering applications (this should be done at the end of your Introduction as well because it will substantially enhance your academic significance, i.e., self-advertising).

References

17. The reference list you provided is inconsistent; e.g., some have the year in brackets ahead while others are at the end with even the month. Please sort those out and get them fixed according to the PloS One standard.

Figures and Quality

18. All figures are provided in very blurry forms, resulting in difficulty in reading and checking all essential details. It could be an issue caused by Word-to-PDF transformation. Moreover, despite Figures 1-5, the font sizes of values in axes, labels, and legends in all other figures are too small to be read; e.g., the axis label “z” is missed in Figure 6 without any unit and interpretation.

19. Last but not least, please scrutinize your manuscript using grammar software (e.g., Grammarly, Writefull, Quiltbot, etc.) to screen out all grammatical mistakes and typos before the subsequent submission of your revised draft.

Reviewer #2: The authors investigated a voxel-based approach for simulating microbial decomposition in soil: comparison with LBM and improvement of morphological models. Before the publication of this paper, I have raised some issues as follows:

Major issues:

1. The authors should improve the Abstract of the study.

2. The authors should revise the Introduction of the study by including the novelty.

3. What was the basis for choosing the mentioned ranges for the investigated parameters?

4. The authors should thoroughly clarify each concept with physical significance.

5. A comparison is expected with the experimental results.

6. When utilizing articles in sentences, authors should be more careful.

7. A reasonable physical explanation should be provided for the observed trends, not only to report what is graphically seen in the figures. More physical insight into the Discussion section is needed.

8. Too many passive sentences are in this article.

9. The conclusions must be More Specific to the obtained results. The present conclusions

are more general.

10. The authors should write the practical applications and advantages of this study.

Minor issues:

Abstract:

Page 1

Line 19: "matter," It seems that an unnecessary comma. Consider removing the comma.

Line 28: "simulation," It appears that you have an unnecessary comma in a compound predicate. Consider removing it.

Line 29: "Mosaic" There seems to be a problem with article usage here.

Introduction :

Page 2

Line 43: "to fully capture its complexity" The to-infinitive to capture has been split by the modifier fully. Avoiding split infinitives can help your writing sound more formal.

Line 45: "microscopy" It seems that you are missing a comma. Consider adding a comma.

Line 49: It appears that starting may be unnecessary in this sentence. Consider removing it

Line 52: It appears that the preposition in is redundant. Consider removing it.

Page 3

Line 58: It seems that LBM based is missing a hyphen. Consider adding the hyphen(s).

Line 60: It seems that the verb require does not agree with the subject. Consider changing the verb form.

Line 61: It seems that real time is missing a hyphen. Consider adding the hyphen(s).

Line 67: The word especially is often overused. Consider using a more specific synonym to improve the sharpness of your writing.

Line 72: "voxels," It seems that you have an unnecessary comma. Consider removing the comma.

Page 3

Line 91: "(100% saturation)," It appears that you have an unnecessary comma in a compound object. Consider removing it.

Page 6

Line 138: The word apply appears repeatedly in this text. Consider using a synonym in its place.

Line 160: "diffusion" It seems that there is an article usage problem here.

Page 7

Line 162: The phrase is equal to maybe wordy. Consider changing the wording.

Page 11

Line 242: The phrase stands out as may be wordy. Consider changing the wording.

Page 12

Line 256: The word respire doesn't seem to fit this context. Consider replacing it with a different one.

Page 15

Line 305: This sentence "as well as" may contain incorrect or awkward phrasing. Consider changing it.

Page 16

Line 331: "then" It seems that you are missing a comma. Consider adding a comma.

Line 332: "S1" It seems that there is an article usage problem here.

Page 17

Line 339: The word involves appears repeatedly in this text. Consider using a synonym in its place.

Line 349: "and" It seems that you are missing a comma. Consider adding a comma.

Page 18

Line 365: It appears that your sentence or clause uses an incorrect form of the verb resulted. Consider changing it.

Line 368: "MATLAB" It seems that there is an article usage problem here.

Page 19

Line 382: "Voxel" It seems that there is an article usage problem here.

Line 385: It appears that you are missing a comma before the coordinating conjunction and in a compound sentence. Consider adding a comma.

Line 386-387: A knowledgeable audience might find this sentence hard to read. Consider breaking it into two.

Line 400: It appears that your sentence or clause uses an incorrect form of the verb distribute. Consider changing it.

Page 20

Line 406: The phrase in accordance with may be wordy. Consider changing the wording.

Line 407: “that we” It seems that there is a pronoun problem here.

Line 417: It appears that results may be unnecessary in this sentence. Consider removing it.

Page 21

Line 427: The word larger is often overused. Consider using a more specific synonym to improve the sharpness of your writing.

Line 427: The phrase illustrates a comparison of may be wordy. Consider changing the wording.

Line 432: “method,” It seems that you have an unnecessary comma. Consider removing the comma.

Line 433: “LBM-based” It seems that there is an article usage problem here.

Line 434: “PNGM-based” It seems that there is an article usage problem here.

Page 22

Line 460-461: It seems that LBM based is missing a hyphen. Consider adding the hyphen(s).

Line 461: “diffusion,” It appears that you have an unnecessary comma in a compound object. Consider removing it.

Line 462: “same” It seems that there is an article usage problem here.

Page 34

Line 679: “stochastic” It seems that preposition use may be incorrect here.

Line 680: “normalized” It seems that there is an article usage problem here.

Line 681: “update” It appears that your sentence or clause uses an incorrect form of the verb update. Consider changing it.

Page 35

Line 697: The phrase compared to may be wordy. Consider changing the wording.

Line 698: The phrase Accurate determination of may be wordy. Consider changing the wording.

Page 36

Line 720: The noun phrase explicit scheme seems to be missing a determiner before it. Consider adding an article.

Line 720: “step,” It appears that you have an unnecessary comma in a compound object. Consider removing it.

Line 726: It appears that from may be unnecessary in this sentence. Consider removing it.

Conclusion:

Page 38

Line 759: The word huge is often overused. Consider using a more specific synonym to improve the sharpness of your writing.

Line 761: “an enhanced” The indefinite article, an, may be redundant when used with the uncountable noun accuracy in your sentence. Consider removing it.

Line 765: It seems that the verb outperform does not agree with the subject. Consider changing the verb form.

Line 769: It appears that hence may be unnecessary in this sentence. Consider removing it.

Reviewer #3: # Summary of the research

The Klai et al. manuscript presents a modeling approach to simulate the microbial decomposition of organic matter in soil that accounts for pore-scale solute transport by diffusion. The approach assumes a two-phase porous medium (solid and liquid). It uses graph theory to abstract the spatial discretization of the pore space based on 3D computed tomography imaging of a soil sample (512^3 voxels with a size of 24 µm^3). The method uses an adjacency matrix to represent a graph constructed from the connectivity of voxels classified as water-filled pore space. Solute transport is simulated using a simplified mass-transfer representation of diffusion. Microbial transformation of organic matter is simulated by numerically solving a differential equation system at each node of the constructed graph. The manuscript aims at improving the accuracy of soil organic matter turnover simulations and advancing a previously developed pore network geometrical modeling method. The new approach is tested against an accurate but computationally demanding reference method (Lattice Boltzmann). The second part of the manuscript applies the introduced approach to improve the computationally efficient pore network modeling approach.

# Overall impression

The presented modeling appraoch has a high potential to contribute in advancing the prediction of microbial decomposition processes in soil while accounting for pore space heterogeneity. However, I see several specific points for improvement. Major weakness of the manuscript is the lack of a comprehensive discussion how the presented work relates to other research in the field. Related to this issue, the authors missed to clearly point out the addresses research challenge and objectives. At the same time part of the conclusion appears partly overreaching.

I think the mathematical presentation of the approach requires substantial improvements. Similarly several critical conceptual points require clarification (see details below). The results and discussion section contains a significant methodological part related to parameter estimation required for the application of the new approach with pore network modeling. While I appreciate the attempt to demonstrate the application of the newly developed modeling approach, this introduces a new story with a second methodology (machine learning) to the manuscript that could be better presented more thoroughly in a separate manuscript. Presenting two weakly related stories in one manuscript leads to a rather long manuscript (38 pages) and a relatively high number of figures (15). In addition some detailed quite technical information could be avoided to make the manuscript more concise.

Overall, I think the manuscript presents an innovative approach but requires substantial reorganization and improvement. With a thorough revision it can provide a substantial methodological contribution to advance simulations of microbial decomposition of organic matter in soils.

# Specific points for improvement

## Abstract

The abstract in principle points out the methodological approach nicely. What I missed was a clear statement on the aim and objectives of the study and also a conclusive statement how this approach advances the research field. So I strongly suggest to clearly point out the research challenge at the beginning. I think the statement on the generality of the model (l 21-22) is not substantiated in the paper and overreaching as the presented approach is limited to a two phase systems. As a minor point, the authors use a number of abbreviations that are not properly defined in a self-contained abstract (LBioS, Mosaic models and LBM in l 25,28).

## Introduction

Current modeling approaches are introduced together with their limitations. However, the authors missed to include a highly relevant and directly related approach (Zech et al. 2022, https://doi.org/10.1016/j.geoderma.2021.115394). This paper coupled the cited approach of Portell et al. (2018) with a cellular automata model. While the approach of Zech et al. (2022) is limited to 2D, it is highly relevant approach which numerically solves a PDE-based reaction-diffussion equation system within the pore space derived from computed tomography imaging. The authors claim in the introduction that their method improves current approaches (l 71-72) and requires no calibration (l 73). Instead of anticipating results in the introduction, clarification is needed to outline which research gap(s) are tackled by what objectives. As in the abstract, the second part of the study is outlined (l 85ff and l 30ff). This part appears as unconnected to the main study which is also evident from the used phrases "Furthermore" (l30) and "In addition" (l 85) in the text.

## Material and methods

### Major issues

The statement on the applicability of the method for unsaturated soil, i.e. to a three-phase system (soilid, liquid, air), needs to be clarified by explaining in more detail what "after a method of drainage" means (l 105-106). Only citing previous paper is insufficent for readers.

Introducing Fick's law (l 142-155) should be significantly shortened as it is textbook knowledge. The derivation of the equations is hard to follow and should be reorganized (section 2.2.2). Throughout the section, the authors mix differential equations with dicretized versions of the equations.

To my understanding, the authors applied an operator splitting approach (c.f l 312-314) to numerically solve a diffusion-reaction partial differential equation (cf. Zech et al. 2022, https://doi.org/10.1016/j.geoderma.2021.115394). If so it should be clearly indicated and derived based on a presentation of the partial differential equation system. In addition, estimates on the accuracy loss due to the chosen numerical approximation scheme should be provided. In my understanding the spatial discretization of the diffusion equation is fixed according to the resolution of the coputed tomography image (24 µm^3), which should be clearly pointed out to readers. The manuscript provides an Eigenvalue decomposition of the spatially disretized ODE system (l 187 - 202), but fails to point out the purpose of this and what the transformed equations are used for.

Section 2.2.3 points out the temporal discretization of eq. 1 using simple implicit and explicit schemes in a high level of detail that is not essential to understand the paper. Thus it could be moved to supplementary material. A major point that arises is that for solving ODE systems much more advanced schemes are available. It is unclear why the authors do not use advanced methods as implemented in state-of-the-art ODE solvers. Also, the authors provide an analytical solution to eq. 1 (l 191) which could have been used directly and it became not clear to me why the equation needed to be solved numerically.

The govering equation describing microbial dynamics (l 270) indicates that growth respiration is not considered, which is in contrast to the fact that microbial catabolisms and anabolism are coupled ,typically represented in microbially explicit models by yield coefficents (c.f. Eq. 1 in Manzoni et al. 2009, https://doi.org/10.1016/j.soilbio.2009.02.031).

It is unclear what "system 3" refers to (l 328) and what's the rationale to apply an asynchonous scheme to solve the reactive ODE system at each graph node (l 329-331) as a coupled ODE system must be solved for all coupled state variables simultaniously to achieve correct results.

The authors claim that the spatial discretization of the model, which is given by the pixel size of 24 µm, is suited to resolve microbialactivity and simulate diffusive transport (l 357-359). It would be important to provide evidence for this claim. However, for simulating diffusion a spatial discretization of 24 µm could be too coarse. In addition, a large part of the pores are not resolved by computed tomography imaging (visible porosity is only 8 and 17%, l 365-366) and pores smaller than 24 µm are potentially relevant for microbial activity. Also, the effective resolution of the image at a pixel size of 24 µm would be rather 50 µm assuming that at least two pixels must be considered to clearly differentiate between water-filled pores and solid matrix. So, it would be important to at least clearly point out the issue of unresolved pore space as a limitation of the approach.

###Minor issues

The equation numbering is incomplete, some equations are numbered, some are not.

Subsection 1.2.3.2 does not exist, this is probably a typo (l 343).

## Results and discussion

In a first part, the section essentially shows the comparison of simulations with the new approach with the other two approaches. In addition, results from numerical tests. A second part introduces an approach and show its results to improve PNGM modeling approaches. I'd recommend to focus on the first part and add a discussion of own results in relation to other experimental and modeling studies in the research field. It is the major weakness of the manuscript that such a discussion is lacking. Another important issue that I see with the results/ discussion section section is that it is overloaded with details on material and methods, which should be shifted in the corresponding section (see details below).

### Major issues

For the pore network geometrical modeling approach a general coefficient alpha=0.6 was used (l 407) and a rationale for selecting this value would be needed for readers.

A specific value for D_DOM was used (l 458-459). A rationale or derivation of the value is not provided, but would be important as this parameter probably affects microbial organic matter turnover by diffusion-limitations of DOC. This includes also the derivation of the unit (voxels^2 j-1). What unit is "j"? I'd expect a diffusion coefficent given in a unit of [length^2]/[time] (e.g., µm^2/h). For me it is also unclear what the authors mean by "with all other diffusion processes of other compounds canceled".

Several tests using explicit and implicit schemes with different time steps are reported in quite some detail (e.g., table 3 and l 479-487). I'd suggest to only report the optimal variant in the main manuscript and move the detailed results to the supplementary material. The presented result take a lot of space (text and figures) in the manuscript but will be not of high interest for most readers. From the presented test results, I also had the impression that the numerical schemes were not the optimal solution. The simple explicit Euler scheme is known to produce low-accuracy results for two large time steps. A time step of 0.1 s was selected as the largest possible (l 483). Could you provide an estimate of the accuracy of the numerical solution? Alternatively more advanced schemes could be used, including adaptive time stepping.

The diffusion conductance coefficients approximation is introduced very detailed but difficult to grasp (section 3.3). Also, it is mostly not a result but a method description. The text uses a notation from machine learning (e.g. training instead of calibration) inconsitantly. For instance, it is indicated that a neural network is used (l 588-589), which is to my understanding not correct and if so more details on the structure of the neural network should be provided.

The stochastic gradient descent method is introduced very detailed (l 611-649), which I would consider as supplemantary information not of high interest for most readers. While the authors describe the method, a reference to the source of the information is missing and would be important to include.

I disagree with the claimed improvement in accuracy of the new PNGM model ( l 722-723). From Figure 15 it is evident that the results from both PNGM model variants are very close and both are not matching well with the VGA simulations.

While section 3.4 indicates a to provide a discussion on limitations of the approach, I could not find it in the text. In contrast, the listed advantages appear overreaching. For instance, the indicated use of neural networks for ubscaling (l 748-752) is rather a potential future approach then a direct outcome of the study (=advantage). Regarding the generality (l 753-754), it would be particularly import to discuss limitations of the presented approach with respect to the representation of microbial community dynamics, spatial discretization (determined by voxel resolution), restriction to a two-phase system and advective transport.

### Minor issues

Missing reference [35] in l 440.

For my understanding two abbreviations are used for the same approach: VGA=GDE, right? Better use only one (e.g. l 510-512).

"more" instead of "much" accurate representation (l 515)?

# Conclusion

The conclusion partly repeats section 3.4. I'd suggest to merge both. What is also missing is a link to other approaches (e.g., Zech et al. 2022, Zech et al. 2022, https://doi.org/10.1016/j.geoderma.2021.115394) to clearly point out where this approach advances the research field.

6. PLOS authors have the option to publish the peer review history of their article (what does this mean? ). If published, this will include your full peer review and any attached files.

**Do you want your identity to be public for this peer review?** For information about this choice, including consent withdrawal, please see our Privacy Policy .

Reviewer #1: No

Reviewer #2: No

Reviewer #3: No

---

## [Author Response · Author response to Decision Letter 1]

2 Sep 2024

Review Comments to the Author

Reviewer #1: This study presents a novel computational method for simulating microbial decomposition in soil using 3D micro-CT images. It uses a valuated graph of connected voxels to model complex diffusion and transformation processes. The method includes parallelization strategies and evaluates various numerical techniques. Validation against LBioS (lattice-Boltzmann method) and Mosaic (Pore Network Geometrical Modelling) shows it matches LBM-based simulation accuracy with a quarter of the computing time and exceeds Mosaic in accuracy without calibration. Additionally, it enhances PNGM-based simulations by approximating diffusional conductance coefficients using stochastic gradient descent.

Based on the initial assessment of this manuscript, the current decision is a major revision to reconsider the publication with Plos One.

Abstract1. Line 18-19: It would be good to point out the research gap before running into aims at the beginning of your abstract (be concise, please; after all, it is just an abstract)

We added a few sentences describing the research gap.

2. Line 25: Could you please define LbioS in your abstract as it first appears in your content?

We defined Lbios within the abstract

3. Line 28: one-fourth -> a quarter

We replaced “one-fourth” by “a quarter” and add a sentence pointed out that the computing time decrease can be a key point.

4.

Line 33: It would be better if you could restate/highlight your engineering contribution from the novel method you developed in this study (please be concise; after all, it is just an abstract).Introduction

The last sentences has been reduced to be more concise but we keep one last sentence dealing with the computation of diffusive conductive coefficients. The reason is that this promising way becomes much more easy to investigate due to the computing time gain.

5. Line 51: These efforts aim to … ? or These studies aim to … ? Can efforts aim to do something?

We replaced “these efforts aim to…” by a more adequate sentence.

6. Line 68: You have not accomplished your literature review by identifying a research gap in high clarity. You have not compared and contrasted the pros and cons from the historical track of previous studies (the initial, later progress, and state of the art by now; what has been done already and what else has not been achieved yet?) in order to identify those research gaps before you propose your research objectives, followed by methods from Line 69. Your review of published works should solidly support your research purposes before heading into it. There is a significant gap or logical disconnection between Lines 68 and 69.

We agree about the need to better highlight the research gap and improve the logical flow between the literature review and our proposed objectives. We have restructured the literature review to more clearly contrast previous studies, identify existing gaps, and establish the rationale for our proposed research objectives and methods

7. Lines 82-87: You have structurally illustrated your paper in the second last paragraph without mentioning the specific sections/chapters. You don’t need to redo it with an independent paragraph that looks slightly redundant.

We agree that the redundancy should be avoided. We have combined the structural overview into a single paragraph and removed the redundant content.

8. At the end of your Introduction and right after your research proposal, you should re-highlight your research significance and engineering contributions to dazzle the reviewers and readers, who have short memories after reading a series of lengthy content, because not all of them can easily recall all the content you provided in the background and literature review in the Introduction. Note that it must be concise and informative.

We have revised the conclusion of the Introduction to re-emphasize our research’s significance and engineering contributions in a concise and impactful manner, ensuring that the key points remain fresh in the reader's mind.

Material and methods

9. Lines 91-92: You can just write, “In this study, we only considered the condition that soil is fully saturated, i.e., soil pore space is filled with water, in exception that soil under the unsaturated condition, i.e., soil void space is partially filled with soil moisture.” An aquatic environment could represent many other meanings, not only in topsoil, vadose zone, and groundwater under the phreatic line (i.e., groundwater table) but also in other water bodies, e.g., nearshore seawater, surface flow, etc.

Thank you for the clarification. We have revised the sentence as suggested to more accurately describe the fully saturated soil condition while avoiding ambiguity regarding the term "aquatic environment."

10. For a numerical contribution like this, I recommend you relocate your pseudo code (algorithm) into an appendix and leave a Matlab or Python code in your supplementary file so many researchers interested in this code will further work on it. It would be much better if you could draw a flow chart for your pseudo-code as a figure in the main text rather than in the appendix and supplementary material.

We appreciate your suggestion. We have replaced the pseudocode with a flowchart, which we believe makes the process more readable and visually intuitive. Additionally, in section 2.5 (Implementation Details), we have included a link (https://github.com/mouadklai/VGA_microbial_decomposition) to the C implementation for further reference.

11. Since it was not observable in your figures in terms of a few very important key points for a numerical simulation, e.g., boundary conditions, initial conditions, etc., you should better visualize those in your simulation demo figures and also provide other information regarding convergence and numerical stability issues (e.g., size of discretization, temporal step in your implicit and explicit schemes, simulating scale compared to single particles or the representativeness of REV, etc.) in the numerical method you just developed. Those will help you defend your work from other peer reviewers who are experts in developing numerical schemes. Results and discussion

Thank you for highlighting the importance of including more details regarding boundary conditions, initial conditions, and key numerical aspects such as discretization size, time-stepping, and representativeness.

Boundary Conditions: These are now detailed in subsection 2.5.3 of the paper.

Initial Conditions: These are explained in the Results and Discussion section, where each simulation for diffusion and microbial activity is compared.

Stability and Convergence: Stability results are discussed in the Results and Discussion section. However, we have added explicit details about numerical stability and convergence in subsection 2.2.3 on numerical schemes for the graph diffusion equation.

Briefly:

Explicit Scheme: Easier to implement and computationally cheaper but requires smaller time steps to maintain stability.

Implicit Scheme: More stable with respect to time step size but involves solving linear systems at each time step.

12. As you used image analysis on CT scans, have you provided adequate technical details on CT scanning and image processing, e.g., voxel-length calibration, ratio, measuring precision, etc.?

Thank you for your valuable comment. We appreciate the need for more information regarding the dataset used for the simulations and how it was processed. we provided additional context and detail regarding the 3D microcomputed tomography (µCT) image data, soil characteristics, and the segmentation techniques applied. It can be found in section 2.6 dataset description and it included a new subsection about boundary conditions

13. Line 516: You claimed you adopted the D3Q9 scheme for LBM simulation. So far, to the best of my knowledge, it should have D2Q9, D3Q15, and D3Q27. Could you please visualize your D3Q9 cell in 3D space, as you demonstrated in Figure 5? Please don’t forget to update all figure numbers in both their captions and content.

Thank you for the feedback. It is true that, by mistake, we referenced the D3Q9 scheme instead of D3Q19. We updated our manuscript accordingly. Below is a figure illustrating the D3Q19 scheme used in this study, sourced from: https://link.springer.com/article/10.1007/s11242-011-9914-7.

14. As for many diffusion coefficients or diffusivities you adopted in your numerical simulations, I recommend you defend your selection of those by comparing them against published works, i.e., necessary citations of new references should be added in. This question could also be raised to select other parameters of the properties of fluid and porous media in your work. It will convince experts who have more experience in this area.

Thank you for your feedback regarding the selection of diffusion coefficients and other parameters. The diffusion coefficients and other biological parameters used in our simulations were adopted from previous works to remain within a similar range of experimentally relevant values. These parameters were chosen based on their practical applicability and relevance to our study, reflecting their experimental nature.

We acknowledge that the exact values of diffusion coefficients can vary; however, our simulations demonstrate that the overall trends and results remain consistent regardless of the specific diffusion coefficient used. This consistency suggests that the main findings are robust and not overly dependent on the precise value of the diffusion coefficient. We clarified this point in the revised manuscript and discuss in section 3 how our results are representative of general behaviour rather than being sensitive to specific parameter values.

15. Lines 734-754: The pros and cons could be better summarized in a table for comprehensive comparison and contrast. You can also do so to compare different numerical methods. It will shorten the content you must deliver to readers and make the paper more concise and informative, i.e., straightforward to read and check all essentials in just a few tables. Conclusions

Thank you for your insightful suggestion regarding the organization of the content in lines 734-754. We agree that summarizing the pros and cons, as well as comparing different numerical methods in a table format, would enhance the readability and conciseness of our paper.

In response to your suggestion, we have included a table that clearly outlines the pros and cons of the different approaches, providing a comprehensive comparison.

16. Please reconstruct your concluding section as follows:Paragraph 1: Concise summary of your research gaps identified through literature review and research objectives; Briefing your methodology with some very critical settings;Paragraph 2: Listing your conclusions or research findings point-by-point;Paragraph 3: Mention the limitations and reflection of your work; Highlight the significance of your research and engineering applications (this should be done at the end of your Introduction as well because it will substantially enhance your academic significance, i.e., self-advertising).References

Thank you for your valuable feedback. We revised the conclusion to address your comments by summarizing the research gaps and objectives, briefly describing the methodology. We then listed the findings and contributions in a point-by-point format, and concluded with a discussion of the limitations and the significance of the research, including its practical applications.

17. The reference list you provided is inconsistent; e.g., some have the year in brackets ahead while others are at the end with even the month. Please sort those out and get them fixed according to the PloS One standard.Figures and Quality

Thank you for pointing out the inconsistencies in the reference list. We have corrected the formatting to align with the PLoS ONE standard.

18. All figures are provided in very blurry forms, resulting in difficulty in reading and checking all essential details. It could be an issue caused by Word-to-PDF transformation. Moreover, despite Figures 1-5, the font sizes of values in axes, labels, and legends in all other figures are too small to be read; e.g., the axis label “z” is missed in Figure 6 without any unit and interpretation.19. Last but not least, please scrutinize your manuscript using grammar software (e.g., Grammarly, Writefull, Quiltbot, etc.) to screen out all grammatical mistakes and typos before the subsequent submission of your revised draft.

Thank you for your feedback regarding the figures. We have changes figures with their best quality. Also, we have corrected figure 6 to include the missing axis label “z”

Reviewer #2: The authors investigated a voxel-based approach for simulating microbial decomposition in soil: comparison with LBM and improvement of morphological models. Before the publication of this paper, I have raised some issues as follows:

Major issues:

The authors should improve the Abstract of the study.

We have updated the abstract to make it clearer and more concise, clearly stating the research challenge, objectives, and contributions of our study.

The authors should revise the Introduction of the study by including the novelty.

We have revised the introduction to clearly highlight the novelty of our approach.

What was the basis for choosing the mentioned ranges for the investigated parameters?

The ranges for the investigated parameters were based on a previous study comparing PNGM and LBM (Monga et al. (2022)). These parameters were selected to match those used in that comparison, ensuring that our focus was on comparing overall simulation results rather than affecting the comparison itself.

4. The authors should thoroughly clarify each concept with physical significance.

We have included detailed explanations of the underlying physical principles.

5. A comparison is expected with the experimental results.

We agree that comparing our results with experimental data is important. This paper primarily aims to evaluate our simulation method against existing approaches. The model used in this study was validated by experimental results from Monga et al. (2014), as cited in subsection 2.3 of the text.

6. When utilizing articles in sentences, authors should be more careful.

Thank you for pointing this out. We have carefully reviewed and revised the use of articles in sentences usage throughout the manuscript.

7. A reasonable physical explanation should be provided for the observed trends, not only to report what is graphically seen in the figures. More physical insight into the Discussion section is needed.

The results now include more physical explanations to provide better context and understanding of the observed trends.

8. Too many passive sentences are in this article.

The manuscript has been revised to reduce passive sentences and improve clarity.

9. The conclusions must be More Specific to the obtained results. The present conclusions are more general.

We have revised the conclusion to be more specific to the obtained results, focusing on the key findings and their implications.

10. The authors should write the practical applications and advantages of this study.

We have a section outlining the practical applications and advantages of our study.

Minor issues:

Abstract:Page 1Line 19: "matter," It seems that an unnecessary comma. Consider removing the comma.

We removed the unnecessary comma after "matter."

Line 28: "simulation," It appears that you have an unnecessary comma in a compound predicate. Consider removing it.

We removed the extra comma in the compound predicate.

Line 29: "Mosaic" There seems to be a problem with article usage here.

We removed the extra comma

Introduction :

Page

---

## [Decision Letter · Decision Letter 1]

16 Sep 2024

PONE-D-24-24748R1A Voxel-based Approach for Simulating Microbial Decomposition in Soil: Comparison with LBM and Improvement of Morphological ModelsPLOS ONE

Dear Dr. Jouini,

Thank you for submitting your manuscript to PLOS ONE. After careful consideration, we feel that it has merit but does not fully meet PLOS ONE’s publication criteria as it currently stands. Therefore, we invite you to submit a revised version of the manuscript that addresses the points raised during the review process.  As noted in Reviewer 3's comments, there are serious concerns about the revised version, with some of the comments not fully addressed. Therefore, you must thoroughly address all of Reviewer 3's comments in your revisedmanuscript.  Please submit your revised manuscript by Oct 31 2024 11:59PM. If you will need more time than this to complete your revisions, please reply to this message or contact the journal office at plosone@plos.org . Please include the following items when submitting your revised manuscript:

We look forward to receiving your revised manuscript.

Kind regards,

Eiyad Ahmad Moh’d Abu-Nada

Academic Editor

PLOS ONE

Reviewers' comments:

Reviewer's Responses to Questions

**Comments to the Author**

1. If the authors have adequately addressed your comments raised in a previous round of review and you feel that this manuscript is now acceptable for publication, you may indicate that here to bypass the “Comments to the Author” section, enter your conflict of interest statement in the “Confidential to Editor” section, and submit your "Accept" recommendation.

Reviewer #1: All comments have been addressed

Reviewer #3: (No Response)

2. Is the manuscript technically sound, and do the data support the conclusions?

Reviewer #1: Yes

Reviewer #3: Yes

3. Has the statistical analysis been performed appropriately and rigorously? 

Reviewer #1: Yes

Reviewer #3: N/A

4. Have the authors made all data underlying the findings in their manuscript fully available?

Reviewer #1: Yes

Reviewer #3: No

5. Is the manuscript presented in an intelligible fashion and written in standard English?

Reviewer #1: Yes

Reviewer #3: Yes

6. Review Comments to the Author

Reviewer #1: The authors have adequately addressed your comments raised in a previous round of review and you feel that this manuscript is now acceptable for publication.

Reviewer #3: I appreciate that author's efforts and think that most of the comments have been addressed adequately. However, changes in the manuscript are unfortunately not indicated everywhere in a way that allows to properly check if the claimed response is taken up in the manuscript (e.g., by providing line numbers of the new manuscript in the response). I found some critical points regarding the abstract where I see a strong discepancy between the claimed improvement and the actual revision of the manuscript. Since the author's responses are not fully trackable in the adapted manuscript for me, I have to consider the provided response to reviewers and the manuscript revision as insufficient and cannot recommend to publish the manuscript in the current version without clarifying the remaining issues.

# Abstract

In their response, the authors claim that aims and objectives are now clearly stated. However, I cannot find such a statement in the abstract and the authors also missed to indicate with specific line numbers which part of the revised text exactly provides aims and objectives of the study. The added sentence "The research gap is to propose forward and backward Euler discretisation scheme to simulate joined diffusion and transformation processes within the context of microbial decomposition" needs to be clarified. Why is it a research gap to propose a discretisation scheme? In my understanding that's a technical detail of the proposed approach but not a research gap.

I could further not find the claimed clarification on the two-phase design of the model.

Additionally, the revised abstract has been further extended and should be significantly shortened by removing details.

# Introduction

At the end of the revised introduction, the authors give a kind second abstract describing the approach and study results in quite some detail (l 99ff in the revised manuscript with track changes). This part should be strongly condensed as it is still anticipating results.

# Material and Methods

The authors state "In the revised manuscript, we have mentioned this point more clearly." Where exactly?

The authors respond the following:

"Moreover, directly using the analytical solution for simulations is complex because it

would need to be coupled with the transformation processes at each simulation step which is not straightforward. While using the analytical solution would be preferable for diffusion alone, its application becomes challenging in our context, as it requires calculating the exponential of a large matrix, introducing additional computational difficulties."

In my understanding this response does not provide a convincing argument. For me and also for readers it remains unclear why calculating the exponentional of a matrix is more computational demanding compared to numerically solving a differential equation.

The rationale for decouping growth from respiration is unfortunately not convincing to me and should be clearly outlined for readers in the manuscript. I checked the Vogel et al. (2015) publication, but could also not find a convincing rationale to decouple growth and growth respiration in that publication.

7. PLOS authors have the option to publish the peer review history of their article (what does this mean? ). If published, this will include your full peer review and any attached files.

**Do you want your identity to be public for this peer review?** For information about this choice, including consent withdrawal, please see our Privacy Policy .

Reviewer #1: No

Reviewer #3: No

---

## [Author Response · Author response to Decision Letter 2]

21 Oct 2024

Reviewers' comments:

Reviewer's Responses to Questions is attached in a separate file named 'Response to Reviewers' as it contains some equations.

---

## [Decision Letter · Decision Letter 2]

1 Nov 2024

A Voxel-based Approach for Simulating Microbial Decomposition in Soil: Comparison with LBM and Improvement of Morphological Models

PONE-D-24-24748R2

Dear Dr. Mohamed Soufiane Jouini,

We’re pleased to inform you that your manuscript has been judged scientifically suitable for publication and will be formally accepted for publication once it meets all outstanding technical requirements.

Kind regards,

Eiyad Ahmad Moh’d Abu-Nada

Academic Editor

PLOS ONE

Additional Editor Comments (optional):

Reviewers' comments:

Reviewer's Responses to Questions

**Comments to the Author**

1. If the authors have adequately addressed your comments raised in a previous round of review and you feel that this manuscript is now acceptable for publication, you may indicate that here to bypass the “Comments to the Author” section, enter your conflict of interest statement in the “Confidential to Editor” section, and submit your "Accept" recommendation.

Reviewer #3: All comments have been addressed

2. Is the manuscript technically sound, and do the data support the conclusions?

Reviewer #3: Yes

3. Has the statistical analysis been performed appropriately and rigorously? 

Reviewer #3: Yes

4. Have the authors made all data underlying the findings in their manuscript fully available?

Reviewer #3: No

5. Is the manuscript presented in an intelligible fashion and written in standard English?

Reviewer #3: Yes

6. Review Comments to the Author

Reviewer #3: While the authors now included links to code and data on GITHUB and kaggle, these repositories seem to be connected to a private account which does not ensure long-term availability to readers. This procedure is not in line with the PLOS Data policy and should be improved such that data provision with the manuscript follows PLOS Data policy.

7. PLOS authors have the option to publish the peer review history of their article (what does this mean? ). If published, this will include your full peer review and any attached files.

**Do you want your identity to be public for this peer review?** For information about this choice, including consent withdrawal, please see our Privacy Policy .

Reviewer #3: No

---

## [Editor Report · Acceptance letter]

PONE-D-24-24748R2

PLOS ONE

Dear Dr. Jouini,

I'm pleased to inform you that your manuscript has been deemed suitable for publication in PLOS ONE. Congratulations! Your manuscript is now being handed over to our production team.

Kind regards,

on behalf of

Prof. Eiyad Ahmad Moh’d Abu-Nada

Academic Editor

PLOS ONE